# Tip60- and sirtuin 2-regulated MARCKS acetylation and phosphorylation are required for diabetic embryopathy

Penghua Yang [1], Cheng Xu[1], E. Albert Reece[1,2], Xi Chen[1], Jianxiang Zhong[1], Min Zhan[3], Deborah J. Stumpo[4], Perry J. Blackshear[4,5] & Peixin Yang[1,2]

Failure of neural tube closure results in severe birth defects and can be induced by high glucose levels resulting from maternal diabetes. MARCKS is required for neural tube closure, but the regulation and of its biological activity and function have remained elusive. Here, we show that high maternal glucose induced MARCKS acetylation at lysine 165 by the acetyl-transferase Tip60, which is a prerequisite for its phosphorylation, whereas Sirtuin 2 (SIRT2) deacetylated MARCKS. Phosphorylated MARCKS dissociates from organelles, leading to mitochondrial abnormalities and endoplasmic reticulum stress. Phosphorylation dead MARCKS (PD-MARCKS) reversed maternal diabetes-induced cellular organelle stress, apoptosis and delayed neurogenesis in the neuroepithelium and ameliorated neural tube defects. Restoring SIRT2 expression in the developing neuroepithelium exerted identical effects as those of PD-MARCKS. Our studies reveal a new regulatory mechanism for MARCKS acetylation and phosphorylation that disrupts neurulation under diabetic conditions by diminishing the cellular organelle protective effect of MARCKS.

[1] Department of Obstetrics, Gynecology & Reproductive Sciences, University of Maryland School of Medicine, Baltimore 21201 MD, USA. [2] Department of Biochemistry & Molecular Biology, University of Maryland School of Medicine, Baltimore 21201 MD, USA. [3] Department of Epidemiology and Public Health, University of Maryland School of Medicine, Baltimore 21201 MD, USA. [4] Signal Transduction Laboratory, National Institute of Environmental Health Sciences, Research Triangle Park, Durham, NC 27709, USA. [5] Departments of Medicine and Biochemistry, Duke University Medical Center, Durham, NC 27710, USA. Correspondence and requests for materials should be addressed to P.Y. (email: pyang@fpi.umaryland.edu)

Neurulation is a process occurring during early embryonic development in which the developing neuroepithelium is folded into the neural tube, which is the primitive form of the central nervous system (CNS). Failed neural tube closure leads to neural tube defects (NTDs), which are severe structural birth defects affecting offspring mortality and morbidity[1–3]. The high glucose (HG) level in maternal diabetes induces NTD formation in both humans and animal models[4–7]. Mitochondrial dysfunction and endoplasmic reticulum (ER) stress in the developing neuroepithelium have been demonstrated to be critically involved in NTD formation in diabetic pregnancies;[8–12] however, the mechanism underlying cellular organelle stress in diabetic embryopathy is unclear. During neurulation, neuroepithelial cells, which are essentially neural stem cells, undergo rigorous proliferation and migration to acquire the competence for neural plate elevation, neural fold convergence extension, and closure. Therefore, neuroepithelial cells may possess mechanisms for the protection of their organelles to ensure cellular homeostasis. Maternal diabetes may disrupt these mechanisms, leading to cellular organelle stress and NTD formation.

Myristoylated Alanine-Rich C Kinase Substrate (MARCKS) is required for neurulation, and deleting the *Marcks* gene results in NTDs, mainly exencephaly[13]. MARCKS protein modification is critical for its biological function. Protein kinase C (PKC) phosphorylates MARCKS, which converts MARCKS from a membrane-bound protein to a cytoplasmic protein[14]. PKCs mediate the cellular stress response because deleting one of the PKC isoforms, i.e., the *Prkcd* gene, abolishes maternal diabetes-induced cellular organelle stress in embryos during the neurulation stage[15]. This evidence suggests that PKC triggers cellular stress by suppressing the protective effects of MARCKS on cellular organelle. However, the predicted cellular organelle protective effect of MARCKS has never been previously demonstrated.

Lysine acetylation is another protein modification that influences the biological action of proteins. The crosstalk between acetylation and phosphorylation in amino acid residues of the same protein has been revealed in transcription factors[16,17]. The acetylation of a transcription factor can positively or negatively regulate the phosphorylation of the same protein, leading to increased or decreased activity of that transcription factor. Although MARCKS is not a transcription factor, it may be regulated by acetylation. In a large proteomic study, the lysine residue lysine 172 of human MARCKS was identified as an acetylation site[18]. Lysine 172 of MARCKS is adjacent to four serine residues that are often modulated by phosphorylation[19]. If MARCKS is regulated by acetylation, it could be interesting to determine whether MARCKS acetylation impacts its phosphorylation.

Since its discovery, histone acetylation has become a well-known euchromatin marker of the activation of gene transcription[20]. Various enzymes involved in histone acetylation, i.e., histone acetyltransferases (HAT), and histone deacetylation (HDAC), i.e., histone deacetylases, have been discovered. Currently, it is recognized that these histone enzymes also acetylate or deacetylate nonhistone proteins[21]. HATs and HDACs are substrate-specific[22]. If MARCKS is acetylated, a specific HAT should pair with an HDAC to regulate MARCKS acetylation. Among the HATs acetylating nonhistone proteins, Tat-interactive protein 60 (Tip60) is activated by cellular stress[23] and induces DNA damage responses[24], which are manifested in diabetes-induced NTDs, suggesting that Tip60 may mediate the cellular stress response in diabetic embryopathy by acetylating protein substrates. In contrast, the seven sirtuin HDAC family members (SIRT1-7) suppress cellular stress by deacetylating nonhistone proteins[17,25]. Therefore, we hypothesize that maternal diabetes induces cellular organelle stress by decreasing SIRT expression, thereby increasing the acetylation of their protein substrates.

Here, we report that MARCKS is regulated by acetylation through Tip60 and SIRT2. We also show that maternal diabetes-induced MARCKS acetylation is required for its phosphorylation, which disables the protective effects of MARCKS on mitochondria and the ER, leading to cellular organelle stress and NTD formation. Preventing the phosphorylation of MARCKS under hyperglycemic conditions either with PD-MARCKS or SIRT2 overexpression restores the protective effects of MARCKS on the neuroepithelium. Thus, we provide mechanistic insight into the essentiality of MARCKS in neurulation.

## Results

**MARCKS acetylation is required for its phosphorylation.** Hyperglycemia in maternal diabetes increases histone acetylation in the developing embryo[26]. Because acetylation in a lysine residue of MARCKS has been detected in human tissues in a global acetylome analysis[18], we sought to determine whether HG in vitro or maternal diabetes in vivo induces MARCKS acetylation. In the mouse neural stem cell line C17.2, HG levels mimic the key adverse effects of maternal diabetes, including cellular stress, apoptosis and histone acetylation, in neuroepithelial cells[9,26–28]. The HG levels triggered MARCKS acetylation in a dose-dependent and time-dependent manner in the C17.2 cells (Fig. 1a, b), whereas mannitol, which is used for the osmotic control of glucose, had no effect on MARCKS acetylation (Supplementary Figure 1A). Consistent with the effect of HG observed in vitro, maternal diabetes increased MARCKS acetylation during the neurulation stage in embryos in vivo (Fig. 1c).

The acetylation site of human MARCKS is lysine 172 (K172)[18]; the site of mouse MARCKS acetylation has not been identified to date. Based on the homologous amino acid sequence alignment between the human and mouse MARCKS sequences, lysine 165 (K165) is the putative acetylation site in mouse MARCKS (Supplementary Figure 1B)[29]. To determine whether HG levels induce MARCKS acetylation in K165, Lysine 165 was mutated into Alanine (A) to prevent acetylation, serving as a dominant negative MARCKS (DN-MARCKS), or into Glutamine (Q) to mimic acetylation, thus serving as a constitutively active form of MARCKS (CA-MARCKS) (Supplementary Figure 1B). The ectopic overexpression of DN-MARCKS blocked the HG-induced acetylation (Fig. 1d), suggesting that DN-MARCKS in the A mutation at K165 competes with endogenous wild-type (WT) MARCKS for acetyltransferase. The blockage of MARCKS acetylation by DN-MARCKS inhibited the HG-induced MARCKS phosphorylation (Fig. 1d). CA-MARCKS mimicked the HG level and induced MARCKS phosphorylation (Fig. 1e). Furthermore, the DN-MARCKS overexpression blocked the constitutively active PKCα-induced MARCKS acetylation and phosphorylation (Fig. 1f). These results suggest that HG levels induce MARCKS acetylation at K165 and that this acetylation is a prerequisite for MARCKS phosphorylation.

**Phosphorylated MARCKS dissociates from the mitochondrion.** Maternal diabetes induces mitochondrial abnormalities in neuroepithelial cells in the developing embryo;[11] however, the underlying mechanism is unclear. Because MARCKS phosphorylation plays a key role in MARCKS cycling from phospholipid membranes, such as the plasma membrane, to the cytoplasm[30], we hypothesize that MARCKS protects neuroepithelial cells from mitochondrial abnormalities and that phosphorylated MARCKS induced by hyperglycemia dissociates from the mitochondrial membrane and does not have a protective effect against the

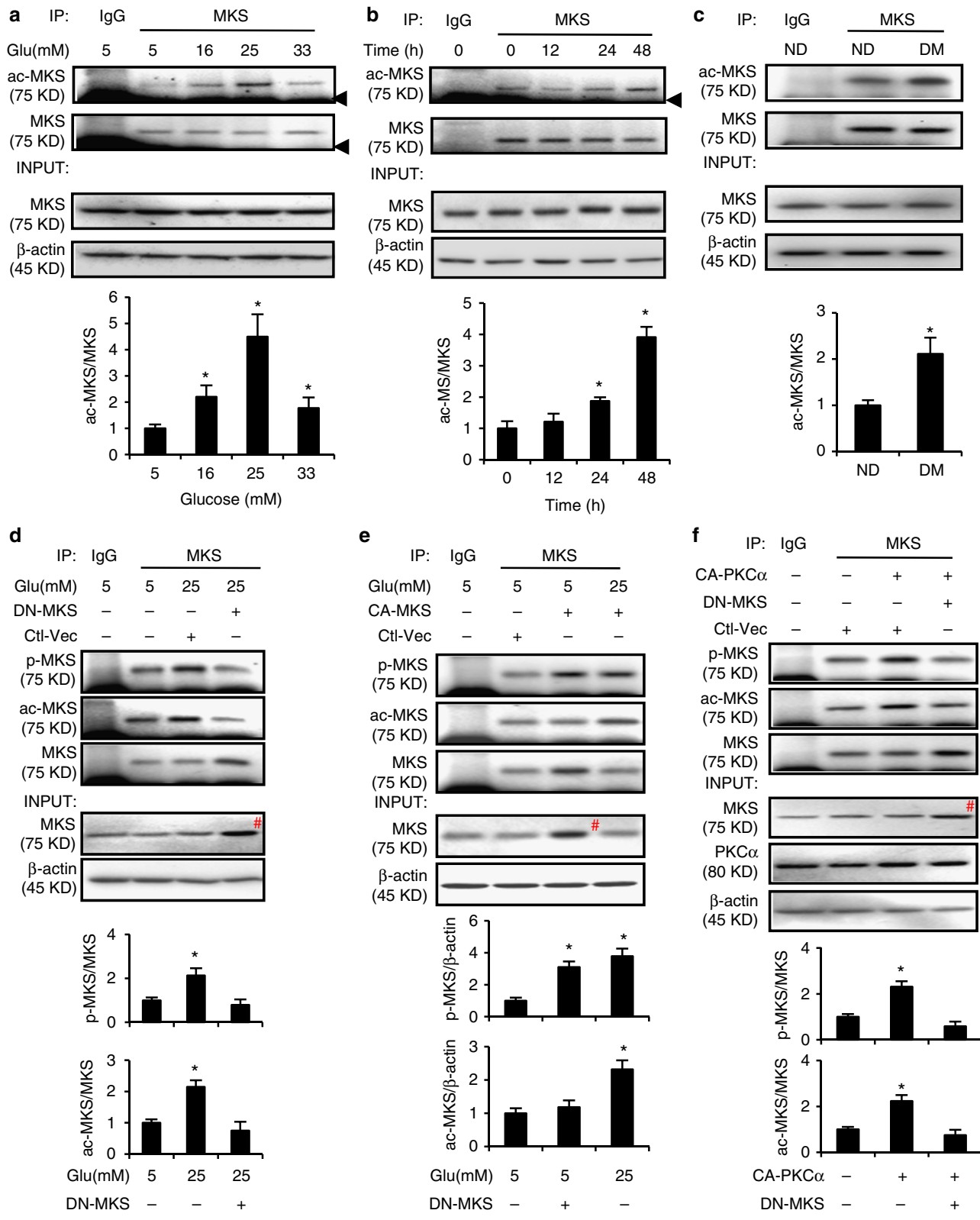

hyperglycemic insult exerted by mitochondria-bound MARCKS on cells. To test this hypothesis, mitochondria were purified from wild-type (WT) and MARCKS-PD (phosphorylation dead) transgenic embryos.

Under nondiabetic conditions, both WT MARCKS and MARCKS-PD were enriched in the mitochondrion (Fig. 2a). Maternal diabetes in vivo or HG in vitro significantly reduced the

amount of MARCKS in the mitochondrial fraction (Fig. 2a, b). The MARCKS-PD overexpression rescued the MARCKS loss in the mitochondria (Fig. 2a, b). p-MARCKS was exclusively located in the cytosolic fractions (Fig. 2a, b). The immunofluorescence staining confirmed that under normal glucose (NG) or non-diabetic conditions, MARCKS was colocalized with the mito-chondrial membrane marker Tom20, whereas p-MARCKS

**Fig. 1** MARCKS acetylation is induced by high glucose and is a prerequisite for its phosphorylation. Abundance of acetylated MARCKS (ac-MKS) in cultured cells (**a**, **b**) and in embryos (**c**). All samples were pulled down by the rabbit anti-MARCKS antibody. Samples from the 5 mM glucose group or the nondiabetic (ND) group were also pulled down by normal rabbit IgG as controls. The incubation time of high glucose was 48 h in (**a**). **d** DN-MARCKS acetylation blocked high glucose-induced MARCKS phosphorylation. **e** CA-MARCKS acetylation increased MARCKS phosphorylation in normal glucose conditions. **f** DN-MARCKS acetylation blocked constitutively active (CA) PKCα-induced phosphorylation. IP: Immunoprecipitation; INPUT: 30 μg proteins from every group were loaded per lane; MKS: MARCKS; NG: normal glucose (5 mM); HG: high glucose (25 mM); DM: diabetes mellitus. All experiments were repeated three times in C17.2 cell line ($N = 3$) in **a**, **b**, **d**, **e**, and **f**. Quantification data were shown in the bar graphs. In C, embryos from three dams per group (Embryos from one dam were needed for one run, $N = 3$) were used. The black arrow heads in **a** and **b** pointed to the IgG bands. # in (**d**–**f**) depicted enhanced MARCKS expression due to MARCKS vector transfection. One way ANOVA with the *Tukey* test was used to analyze the data of panel **a**, **b**, **d**, **e** and **f**. *t* test was used to analyze the data of panel **c**.* Indicates significant difference ($P < 0.05$) compared to other groups

induced by HG or maternal diabetes did not overlap with Tom20 (Fig. 2c–e). MARCKS-PD diminished the HG-induced or maternal diabetes-induced p-MARCKS signals and restored the colocalization of MARCKS and Tom20 (Fig. 2c–e). The proximity ligation assay confirmed the interaction between MARCKS and Tom20 under the nondiabetic or NG conditions (Fig. 2f, g), whereas MARCKS-PD restored the interaction between MARCKS and Tom20 disrupted by maternal diabetes or HG (Fig. 2f, g).

To determine the effect of MARCKS-PD on maternal diabetes-induced or HG-induced mitochondrial abnormalities, the structures of the mitochondria were examined by electronic microscopy. Maternal diabetes significantly increased the number of defective mitochondria (Fig. 3a), which is consistent with our previous findings[11]. Under the diabetic conditions, the overexpression of MARCKS-PD significantly reduced the number of defective mitochondria (Fig. 3a). The activation of pro-apoptotic members of the Bcl-2 family indicates mitochondrial abnormalities. The mitochondrial enrichment in Puma, Bim, Bak, and Bik was increased in the embryos exposed to maternal diabetes (Fig. 3b, c). The increased mitochondrial translocation of these four Bcl-2 family members by maternal diabetes was abrogated in the embryos with the MARCKS-PD overexpression (Fig. 3b, c). Similarly, MARCKS-PD inhibited the HG in vitro-induced mitochondrial translocation of these four Bcl-2 family members (Fig. 3d, e). Because MARCKS is predominantly expressed in neural tissues[31,32], these findings collectively suggest that mitochondrial abnormalities induced by maternal diabetes occurs in the neuroepithelium and that MARCKS protects the neuroepithelium from mitochondrial abnormalities.

**Phosphorylated MARCKS detaches from the ER.** ER stress is another form of cellular organelle stress involved in the pathogenesis of diabetic embryopathy[9,12]. We hypothesize that MARCKS is present in the ER membrane and that its dissociation from the ER membrane after phosphorylation caused by maternal diabetes leads to ER stress. The colocalization signal of calnexin, which is an ER membrane marker, and MARCKS in the neuroepithelia of embryos from diabetic dams was significantly lower than that in the embryos from the nondiabetic dams (Fig. 4a). MARCKS-PD overexpression in the neuroepithelium reversed the maternal diabetes-suppressed calnexin-MARCKS colocalization (Fig. 4a). HG levels in vitro also reduced MARCKS localization in the ER, and MARCKS-PD restored the calnexin-MARCKS colocalization under the HG conditions (Fig. 4b). The proximity ligation assay confirmed that MARCKS binds the ER membrane under NG or nondiabetic conditions (Fig. 4c, d). MARCKS-PD prevented the dissociation of MARCKS from the ER induced by HG in vitro and maternal diabetes in vivo (Fig. 4c, d).

To further investigate whether maternal diabetes-disrupted MARCKS localization in the ER causes ER stress in the developing embryo, the unfolded protein response (UPR) pathways IRE1α and PERK were evaluated. The phosphorylation of IRE1α and JNK1/2 was induced by maternal diabetes. MARCKS-PD diminished the phosphorylation of these molecules (Fig. 4e, f). XBP1, which is a downstream effector of IRE1α, was also activated by maternal diabetes and inactivated in the MARCKS-PD transgenic embryos (Fig. 4g). The PERK pathway, including PERK, eIF2α, and CHOP, was activated by maternal diabetes, and MARCKS-PD abrogated the activation of the PERK pathway (Fig. 4h). Similarly, the HG levels activated the IRE1α and PERK pathways, which were suppressed by the MARCKS-PD overexpression in vitro (Supplementary Figure 2A, B). These results indicate that MARCKS can regulate the ER activity status through phosphorylation and that maternal diabetes-induced MARCKS phosphorylation leads to ER stress.

**MARCKS-PD reduces maternal diabetes-induced NTD formation.** Subsequently, we assessed the functionality of maternal diabetes-induced MARCKS phosphorylation using a transgenic (Tg) mouse line with MARCKS-PD overexpression in the developing neuroepithelium (Fig. 5a). As expected, maternal diabetes increased the phosphorylation of MARCKS in the wild-type embryos (Fig. 5b). However, the phosphorylation of MARCKS was diminished in the MARCKS-PD overexpressing embryos from the diabetic dams (Fig. 5b). The MARCKS phosphorylation impeded its cellular organelle protective effect and led to mitochondrial abnormalities and ER stress, which are critically involved in maternal diabetes-induced embryonic anomalies[11,33]. Thus, we examined the effect of MARCKS-PD on NTD formation. The NTD rate in the MARCKS-PD overexpressing embryos was significantly lower than that in the wild-type embryos from the diabetic dams and was comparable to that in the embryos from the nondiabetic dams (Fig. 5c, d and Supplementary Table 1). Thus, MARCKS-PD ameliorated maternal diabetes-induced NTDs. Caspase cleavage-triggered neuroepithelial cell apoptosis is observed in diabetes-induced NTDs[27,33]. MARCKS-PD ablated caspase 3 and caspase 8 cleavage in embryos exposed to diabetes (Fig. 5e, f) and reduced the number of apoptotic neuroepithelial cells in embryos from diabetic dams compared to that in embryos from nondiabetic dams (Fig. 5g).

NTDs can result from genetic insults that cause premature neurogenesis[34]. However, our previous study demonstrated that maternal diabetes delays neurogenesis in the developing neuroepithelium as evidenced by the absence of the expression of TuJ1 (class III β tubulin), which is a marker of terminal neuronal differentiation, from E8.75 to E9.0 and the enhanced expression of Sox1, which is a marker of neural stem cells[11]. Consistent with our previous findings, TuJ1-expressing neurons were absent, whereas Sox1 expression was enhanced in the E8.75 neuroepithelia from wild-type embryos exposed to maternal diabetes (Fig. 5h, i). In contrast, TuJ1-expressing neurons were present in the floor plate of the E8.75 MARCKS PD transgenic embryos from the diabetic dams, and the expression of Sox1 was lower than that in the wild-type embryos from the diabetic dams (Fig. 5h, i). These results suggest that MARCKS PD

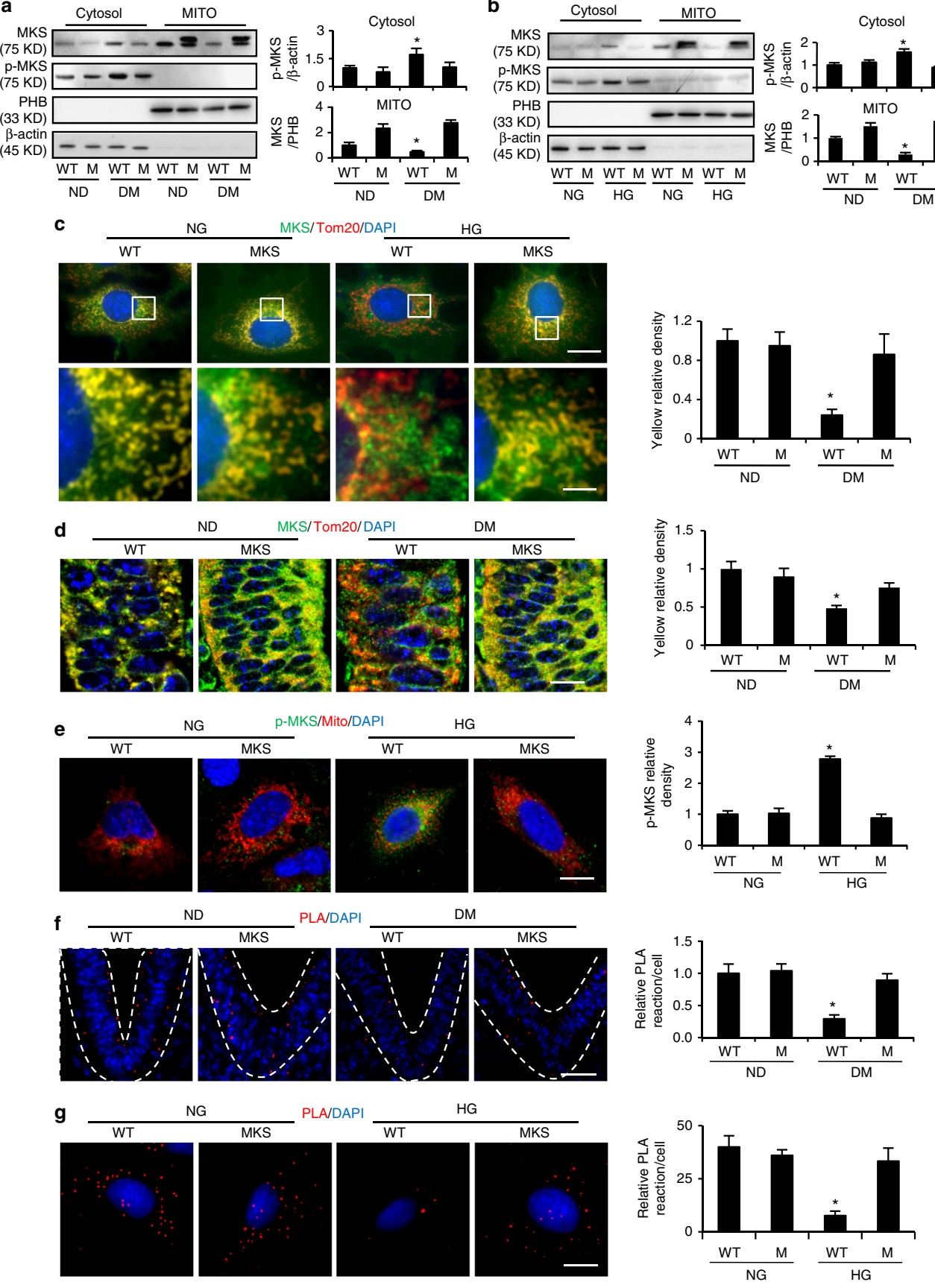

**Fig. 2** Phosphorylated-MARCKS dissociation from mitochondrial membrane. Cytosol and mitochondrial MARCKS and p-MARCKS in E8.5 embryos (**a**) and cultured cells (**b**). Bar graphs were quantification data. M: MARCKS-PD; MKS: MARCKS; PHB: prohibitin. (**c**) Co-localization of MARCKS (green) with the mitochondrial membrane protein, Tom20 (red), in cultured cells. The bar graph was the quantification data of yellow density (co-localization). The region with white squire square in upper panel were amplified in lower panel. Bar in upper and lower panel indicates 17.5 μM and 7.5 μM respectively, **d** Co-localization of MARCKS (green) with the mitochondrial membrane protein, Tom20 (red), in the developing neuroepithelium of E8.75 embryos. The bar graph was the quantification data of yellow density (co-localization). Bar indicates 12.5 μM. **e** Localization of phosphorylated MARCKS (green) in the cytoplasm of cultured cells. Mitochondria membrane (red) was labeled by transfection of a plasmid (DsRed2-Mito-7, Addgene, #55838). The bar graph was the quantification data of green density. Bar indicates 15 μM. Co-localization of MARCKS and Tom20 was assessed by the proximity ligation assay (PLA) in the embryonic neuroepithelium (**f**) (Bar indicates 40 μM) and cultured cells (**g**) (Bar indicates 15 μM). Dot lines indicates neural tube. Red dots indicated that MARCKS interacted with Tom20. The bar graph was the quantification data of red PLA dots per cell. ND nondiabetic, DM diabetes mellitus, NG normal glucose (5 mM), HG high glucose (25 mM). All experiments were repeated three times. In A, three litters per group (one litter per run) were used. One way ANOVA with the *Tukey* test was used to analyze the data. * Indicates significant difference ($P < 0.05$) compared to other groups

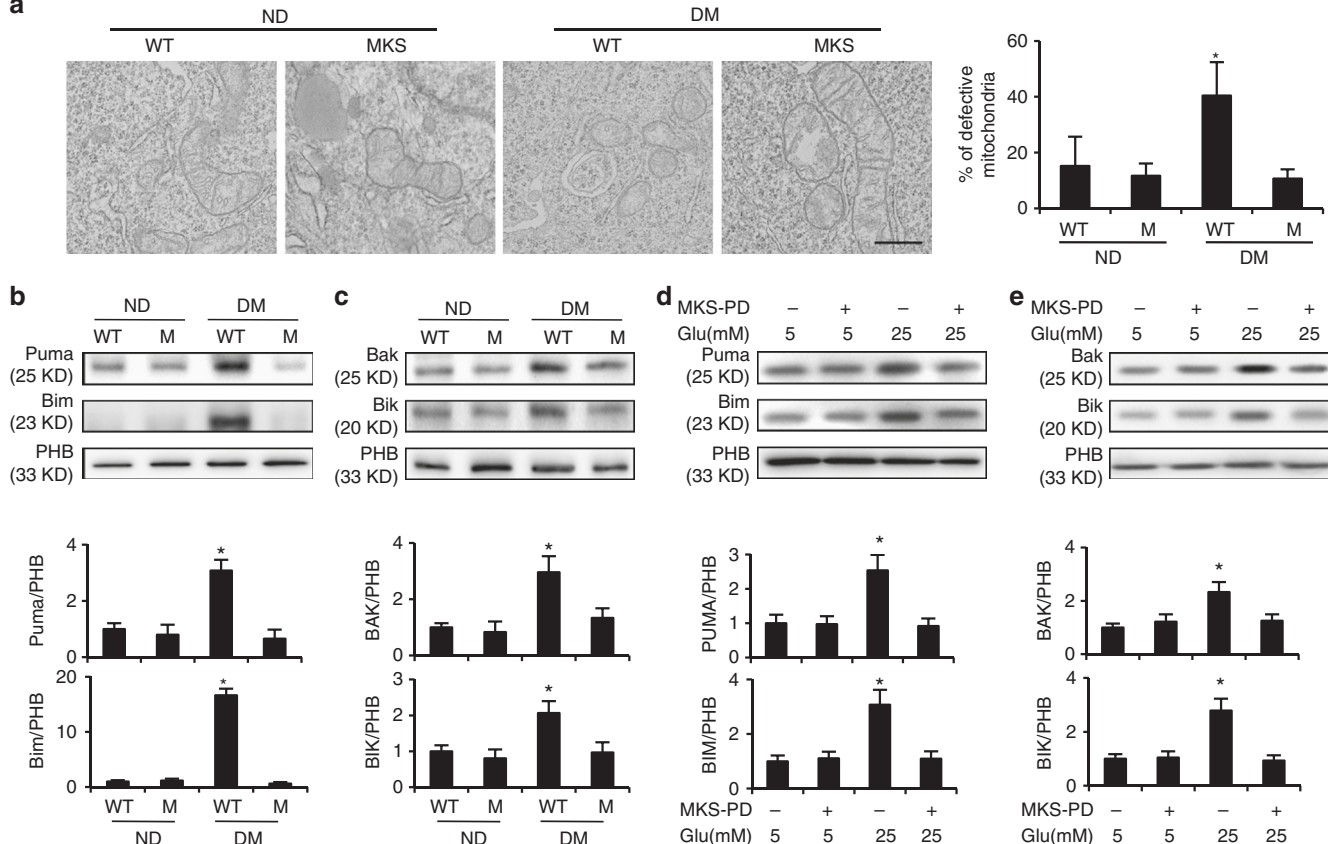

**Fig. 3** MARCKS dissociation from mitochondria leads to mitochondrial abnormalities. **a** Defective mitochondria in the developing neuroepithelium examined by electron microscope. The bar graph was the quantification data of the percentage of defective mitochondria: the number of defective mitochondria number divided by the number of total mitochondria in a given area. Bar indicates 1 μM. **b** Mitochondrial abundance of Puma and Bim in embryos. **c** Abundance of Bak and Bik in the mitochondria isolated from embryos. Quantification data was shown in the bar graph. **d**, **e** Abundance of Puma, Bim, Bak and Bik in the mitochondria isolated from cultured cells. MKS MARCKS, M: MARCKS-PD, ND: nondiabetic, DM: diabetes mellitus. NG: normal glucose (5 mM), HG: high glucose (25 mM). All experiments were repeated three times. In **b** and **c**, one embryo from one dam in one group were performed for one run. Each experiment was repeated three times with three embryos ($N = 3$) from three different dams in one group. All quantification data were indicated as means ± standard derivation. One way ANOVA with the *Tukey* test was used to analyze the data. * Indicates significant difference ($P < 0.05$) compared to other groups

overexpression in the neuroepithelium restores maternal diabetes-delayed neurogenesis.

**Tip60 interacts with and acetylates MARCKS**. MARCKS acetylation was prerequisite to its phosphorylation (Fig. 1d–f). Subsequently, we sought to identify the acetyltransferase responsible for MARCKS acetylation. Among the histone/protein acetyltransferases that acetylate nonhistone proteins, Tip60 is activated

by cellular stress and induces DNA damage responses[24], which are manifested in diabetes-induced NTDs[27,33], suggesting that Tip60 may mediate the cellular stress response in diabetic embryopathy by acetylating protein substrates. A co-immunoprecipitation (CoIP) assay was performed to evaluate the interaction between MARCKS and the three most common protein acetyltransferases, i.e., Tip60, Gcn5, and p300, using embryonic tissue extracts. Maternal diabetes enhanced the interaction between MARCKS and Tip60, whereas MARCKS-

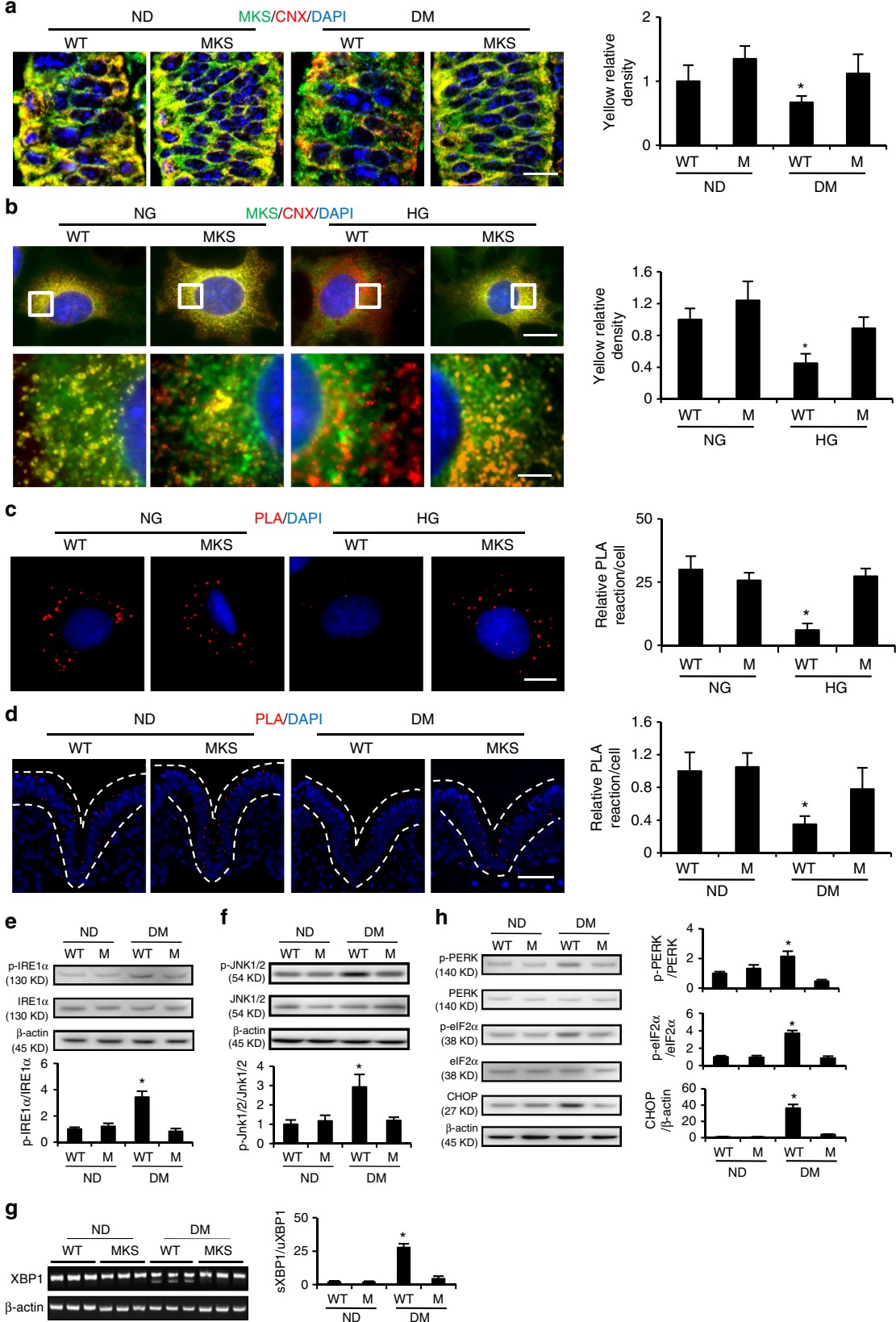

Gcn5 and MARCKS-p300 complexes were not observed (Fig. 6a, Supplementary Figure 3A). Similarly, HG levels increased the formation of MARCKS-Tip60 complexes (Supplementary Figure 3B). These data suggest that Tip60 may be responsible for MARCKS acetylation. The Tip60 siRNA knockdown abrogated

the HG-induced MARCKS acetylation (Fig. 6b). Moreover, ectopic Tip60 overexpression mimicked the HG levels and induced MARCKS acetylation (Fig. 6c). Finally, recombinant Tip60 directly acetylated recombinant MARCKS in a noncell system (Fig. 6d, e). Thus, Tip60 is a MARCKS acetyltransferase.

**Fig. 4** Maternal diabetes dissociates MARCKS from the ER membrane and induces ER stress. Co-localization of MARCKS (green) with the ER membrane protein, calnexin CNX (red), in the embryonic neuroepithelium (**a**) (Bar indicates 12.5 μM) and cultured cells (**b**) (Bar in upper and lower panel indicates 17.5 μM and 7.5 μM respectively). The region with white squire square in upper panel were amplified in lower panel. The bar graphs were the quantification data of yellow density (co-localization). **c** Co-localization of MARCKS and IRE1α was assessed by the proximity ligation assay (PLA) in cultured cells. Red dots indicated that MARCKS interacted with IRE1α. The bar graph was the quantification data in red PLA dots per cell. Bar indicates 15 μM. **d** Co-localization of MARCKS and the ER membrane protein, IRE1α, was assessed by the proximity ligation assay (PLA) in the embryonic neuroepithelium. Red dots indicated that MARCKS interacted with IRE1α. Dot lines indicates neural tube. The bar graph was the quantification data of red PLA dots per cell. **e** p-IRE1α abundance in embryos. Bar indicates 70 μM. **f** Abundance of p-JNK1/2 in embryos. Quantification data was shown in the bar graph. **g** XBP1 splicing in embryos. The bar graph was the quantification data for the ratio of spliced-XBP1 (sXBP1) to unspliced-XBP1 (uXBP1). **h** Protein abundance of p-PERK, eIF2α and CHOP in embryos. ND nondiabetic, DM diabetes mellitus, NG normal glucose (5 mM), HG high glucose (25 mM). All experiments were repeated three times. Three embryos from three different dams ($N = 3$) were performed in panel **e**, **f**, **g** and **h**. One way ANOVA with the *Tukey* test was used to analyze the data. * Indicates significant difference ($P < 0.05$)

**SIRT2 interacts with and deacetylates MARCKS.** The following question is which deacetylase targets MARCKS. The sirtuin (SIRT) deacetylase family includes seven members (SIRT1-7) and regulates an array of cellular functions by primarily suppressing cellular stress[35]. SIRT1-7 deacetylates both histone and non-histone proteins[36]. We recently found that maternal diabetes in vivo and HG in vitro significantly reduced SIRT2 expression[26], suggesting that SIRT2 may play an important role in the regulation of MARCKS acetylation. Maternal diabetes in vivo and HG in vitro significantly reduced the interaction between SIRT2 and MARCKS (Fig. 6f, Supplementary Figure 3C). The ectopic SIRT2 overexpression in cultured cells attenuated the HG-induced MARCKS acetylation (Fig. 6g). The SIRT2 siRNA knockdown in vitro increased MARCKS acetylation (Fig. 6h). The recombinant SIRT2 directly deacetylated recombinant MARCKS, which was acetylated by Tip60, and endogenous MARCKS derived from cells treated with HG (Fig. 6i, j). Deleting the *Sirt2* gene in vivo increased MARCKS acetylation (Fig. 6k). These data collectively support our hypothesis that SIRT2 deacetylases MARCKS.

SIRT2 KO embryos did not exhibit any NTDs under the nondiabetic conditions (Supplementary Table 2), suggesting that MARCKS acetylation only is not sufficient to induce NTDs. To determine whether a pathogenic maternal diet could lead to NTD formation in the SIRT2 KO embryos, SIRT2 heterozygous (SIRT2$^{+/-}$) female mice were fed a pathogenic maternal diet[37] and mated with SIRT2$^{+/-}$ male mice. The wild-type (WT), SIRT2$^{+/-}$, and SIRT2$^{-/-}$ embryos all developed normally without NTDs (Supplementary Table 2), suggesting that even though SIRT2 KO increases MARCKS acetylation, a pathogenic maternal diet might not be able to induce MARCKS phosphorylation, which is a requirement for NTD formation. Because SIRT2 downregulation and PKCα activation are concomitantly present in diabetic embryopathy[2,3], we tested whether PKCα activation in combination with SIRT2 KO could result in NTDs. WT, SIRT2$^{+/-}$, and SIRT2$^{-/-}$ embryos were cultured with or without ROPA, which is a PKCα activator. While the ROPA treatment did not induce NTDs in the WT embryos, this treatment induced 16.7% and 55.6% NTDs in the SIRT2$^{+/-}$ and SIRT2$^{-/-}$ embryos, respectively (Fig. 6l, Supplementary Table 3). Increased MARCKS acetylation was observed in the SIRT2$^{-/-}$ embryos (Fig. 6m), and the ROPA treatment increased MARCKS phosphorylation in the SIRT2$^{-/-}$ embryos compared to that in the WT embryos (Fig. 6n). This evidence supports our hypothesis that both MARCKS acetylation and phosphorylation are required for NTD induction and that MARCKS acetylation enables its phosphorylation by PKCα.

**SIRT2 alleviates maternal diabetes-induced NTDs.** Because we observed that SIRT2 negatively regulates MARCKS acetylation, we tested the functionality of SIRT2 overexpression in vivo. We created a transgenic mouse line in which the SIRT2 transgene was

driven by the promoter of the neuroepithelial cell marker nestin[27]. Thus, in this transgenic mouse line, SIRT2 was specifically overexpressed in the developing neuroepithelium (Fig. 7a). The SIRT2 mRNA and protein abundance in the transgenic embryos was increased by approximately 5.5-fold and 2.3-fold, respectively (Fig. 7b, c), which is equivalent to the magnitude of the SIRT2 reduction by maternal diabetes[26]. Additionally, the *Sirt2* transgene restored the embryonic SIRT2 protein abundance suppressed by maternal diabetes (Fig. 7c). The SIRT2 overexpression in the developing neuroepithelium significantly decreased MARCKS acetylation in the embryos from the diabetic dams (Fig. 7d).

To determine the effect of restoring SIRT2 expression in the neuroepithelium on diabetes-induced NTDs, SIRT2 transgenic male mice were crossed with wild-type female mice (Supplementary Table 4). Maternal diabetes significantly increased the NTD rate in the wild-type embryos (Fig. 7e, f and Supplementary Table 4). Restoring SIRT2 expression in the neuroepithelium ameliorated NTD formation in the diabetic pregnancies (Fig. 7e, f and Supplementary Table 4). In addition, the *Sirt2* transgene restored mitochondrial function and suppressed ER stress in embryos from diabetic dams (Fig. 7g–i). Cellular organelle stress leads to cell apoptosis in the developing neuroepithelium[33]. Restoring SIRT2 expression blocked maternal diabetes-induced caspase 3 and 8 cleavage and neuroepithelial cell apoptosis (Fig. 7j–l). These finding suggests that SIRT2 functions as a MARCKS deacetylase and helps protect embryos from maternal hyperglycemic insults.

In summary, MARCKS acetylation, which is reversibly regulated by Tip60 and SIRT2, is a prerequisite for MARCKS phosphorylation and mediates the teratogenicity of maternal diabetes in NTD induction by causing cellular organelle stress. MARCKS is one of the more than 300 genes required for neurulation[38]. However, information about the function of these genes during neurulation is limited. Our study provides an example elucidating the function of these neural tube closure essential genes with the ultimate goal of revealing the cause of NTDs. Additionally, MARCKS is broadly involved in an array of neurological disorders[39], and identifying inhibitors of MARCKS acetylation and phosphorylation is fundamentally important for preventing diabetes-induced NTDs and other neurological disorders.

**Discussion**

MARCKS was originally discovered as a major substrate of PKC in the CNS[40]. Subsequent studies have revealed that MARCKS plays a vital role in neurulation. MARCKS knockout mice exhibit a spectrum of neural defects, including exencephaly, omphalocele, and forebrain commissures, and failure of fusion of the cerebral hemispheres[13]. Recent studies have attempted to unravel the functionality of MARCKS in CNS development. MARCKS has

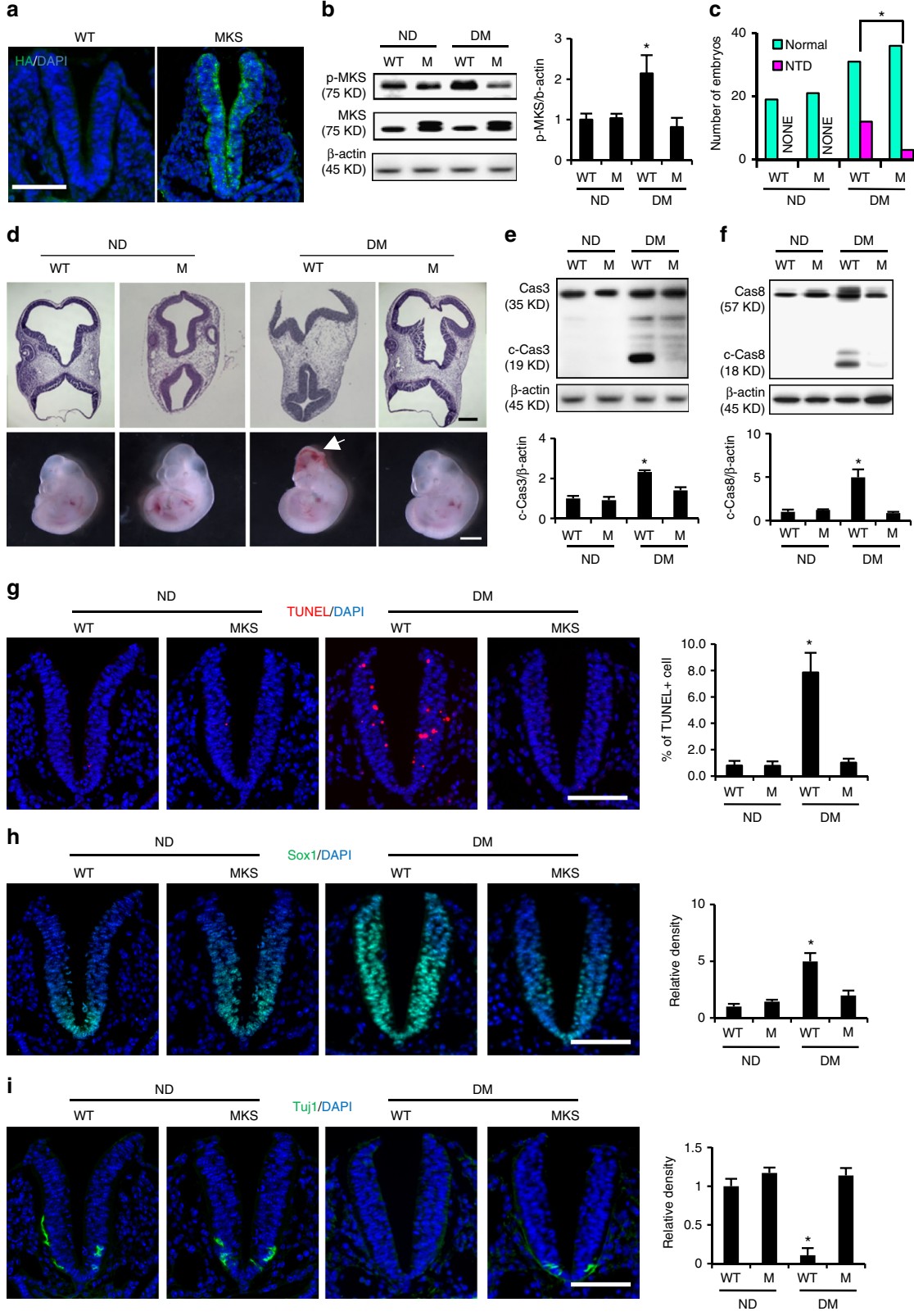

been demonstrated to be involved in neurite initiation and outgrowth[41], modulate radial glial placement and expansion[42], and promote axon development[43]. Abnormalities induced by MARCKS deficiency can be rescued, at least partially, by the following two types of MARCKS mutants: the phosphorylation-dead site mutant MARCKS-PD (serines to asparagines)[19] and the nonmyristoylated MARCKS mutant[44,45]. These findings suggest that membrane-associated MARCKS is required for, while cytoplasmic phosphorylated MARCKS may be detrimental to, CNS development.

Under diabetic conditions, hyperglycemia activates PKCα and PKCδ, leading to NTD formation in the developing embryo[27,46–49]. Both PKCα and PKCδ can induce MARCKS phosphorylation[50]. Increased MARCKS phosphorylation is implicated in the etiology of

**Fig. 5** MARCKS-PD attenuates maternal diabetes-induced neural tube defects. **a** MARCKS-PD overexpression (green signals) was observed only in the neuroepithelia of transgenic embryos. Bar indicates 70 μM. **b** Abundance of phosphorylated MARCKS. Quantification data was shown in the bar graph. **c** Numbers of normal and NTD embryos. * indicates significant difference (P < 0.05) in a Chi-square test. **d** View of closed or open neural tubes from coronal sections of normal and NTD (white arrow) embryos. Bar in upper and lower panel indicates 600 μM and 1 mm respectively. **e**, **f** Abundance of cleaved caspase 3 and 8. **g** Apoptotic cells labeled as red in embryonic neuroepithelia. Representative images of immunofluorescent staining for the neural stem cell marker Sox1 (green). Bar indicates 70 μM. (**h**) and the neuron marker Tuj1 (green) (**i**). Bar indicates 70 μM. ND nondiabetic, DM diabetes mellitus, M: MARCKS-PD. Quantification data was shown in the bar graphs. Three embryos from three different dams (N = 3) were performed in **b**, **e** and **f**. Sample sizes in (**c**) were shown in Supplementary Table 1. One way ANOVA with the Tukey test was used to analyze the data. * Indicates significant difference (P < 0.05) compared to other groups

Alzheimer's disease (AD)[51], which is another neurological disorder. Amyloid beta, which is the toxic species responsible for AD pathology, causes MARCKS phosphorylation through PKC[51]. Our original hypothesis was that MARCKS phosphorylation is the pathological cause of the failure of neural tube closure in embryos derived from diabetic mothers. We observed that NTDs induced by maternal diabetes are inhibited by MARCKS-PD, supporting the causal role of MARCKS phosphorylation in failed neural tube closure in diabetic pregnancies.

MARCKS is associated with the cell plasma membrane and the intracellular membranes of mucin granules[52]. MARCKS binds the membranes of mitochondria and the ER, and the phosphorylation of MARCKS caused by hyperglycemia causes it to dissociate from these intracellular membranes in embryos during the neurulation stage. Indeed, one proteomic study revealed enrichment in MARCKS in the mitochondrial fraction of neurons[53]. Once MARCKS binds the plasma membrane, it increases the abundance of the phospholipid phosphatidyl-inositol (4,5) bisphosphate in the plasma membrane[54]. The five phenylalanines in the MARCKS effector domain, in which the four PKC-phosphorylatable serines also reside, insert into the plasma membrane by further stabilizing the membrane[54].

MARCKS likely maintains the integrity of and protects cellular organelle membranes from stress. During the neurulation stage, cellular organelles in embryos have been shown to be vulnerable to stress insults[11,55]. Phosphorylated MARCKS mimics the effect of MARCKS deficiency[56,57]. MARCKS phosphorylation leads to the loss of its function in protecting vascular endothelial cells from oxidative stress[56]. Thus, phosphorylated MARCKS is an inactive form of MARCKS. MARCKS phosphorylation under diabetic conditions abrogates its protection of cellular organelles.

MARCKS deficiency and phosphorylation mediate oxidative stress-induced vascular endothelial cell abnormalities[56,57]. Non-phosphorylatable MARCKS abolishes, whereas pseudophosphorylated MARCKS mimics, PKC-induced neuron dysfunction[58]. This evidence suggests that MARCKS protects cells from stress, and its deficiency or phosphorylation causes cellular stress. Maternal diabetes-phosphorylated MARCKS dissociates from the mitochondrial membrane and the ER, leading to mitochondrial abnormalities and ER stress, which are two causal events in diabetic embryopathy[11,27].

Mitochondria are essential for embryogenesis[59]. Both the inadequate function of mitochondrial folate metabolism and the lack of key mitochondrial redox genes result in NTDs[60,61]. The overexpression of the mitochondrial antioxidant superoxide dismutase 2 ameliorates maternal diabetes-induced NTDs[62]. Here, we demonstrate that mitochondrial abnormalities due to MARCKS phosphorylation causes NTDs in diabetic pregnancies. MARCKS-PD blocks MARCKS phosphorylation-induced mitochondrial abnormalities by binding the mitochondrion and ER, leading to NTD reduction under diabetic conditions.

Normal ER function is required for neural tube closure. ER stress activates UPR signaling. The active form of XBP1, which is a major downstream effector of IRE1α, induces NTDs in

Xenopus[63,64], suggesting that the IRE1α-ER stress pathway is highly relevant to diabetes-induced NTDs. ER stress is present in the neuroepithelia of embryos exposed to maternal diabetes, and the ER stress inhibitor 4-phenylbutyric acid reduces HG-induced NTDs in cultured embryos[11]. We showed that MARCKS-PD inhibits maternal diabetes-induced ER stress, supporting the protective effect of unphosphorylatable MARCKS on the ER.

MARCKS acetylation is required for its phosphorylation. The inhibition of MARCKS acetylation by SIRT2 blocks maternal diabetes-induced ER stress and NTD formation. Thus, maternal diabetes-induced MARCKS acetylation and subsequent phosphorylation disrupt the association between MARCKS and the ER and mitochondrion, leading to cellular organelle stress.

The phosphorylation and myristylation of MARCKS are well-recognized post-translational modifications that are primarily involved in the binding of MARCKS to or release from plasma membranes[39]. The four serine residues in the effector domain of MARCKS, which have negative charges, bind the negatively charged phospholipids of the plasma membrane[65]. The N-terminal myristoyl group of MARCKS acts second by binding to the plasma membrane and facilitating a relatively stable interaction[65].

MARCKS on lysine residue 165 (K165) is important for its phosphorylation by PKC in neural cells. The phosphorylation sites in MARCKS PSD (phosphorylation site domain) are Serine 152 (S152), S156, S160 and S163. The acetylation residue K165 is adjacent to these four serine residues. K165 acetylation introduces another negative charge that may neutralize the positive charge in the nearby amino acid residues, leading to a MARCKS protein with a more open structure, particularly in the PSD. This structural change may render PKC more accessible to the four serine sites and potentiate these sites to becoming phosphorylated. However, directly investigating the physical function of acetylation on phosphorylation is very challenging. Future studies may explore these structural changes in the MARCKS protein upon acetylation.

We revealed that acetylation, which is another form of post-translational modification, regulates MARCKS phosphorylation under HG conditions. Protein function and subcellular localization are regulated by phosphorylation and acetylation[66]. Phosphorylation plays a key role in cellular signaling, whereas acetylation functions in regulating protein expression and improving protein stability[66]. In some cases, the acetylation of proteins, especially transcription factors, enhances their transcriptional activity by increasing their DNA binding affinity, protein stability and phosphorylation sensitivity[67]. Similar to proteins, which become more sensitive to phosphorylation after they are first acetylated, acetylation is a prerequisite for MARCKS phosphorylation. Indeed, the MARCKS acetylation constitutive active isoform induces its phosphorylation under NG conditions. Moreover, the MARCKS acetylation dominant negative isoform blocks MARCKS phosphorylation increased by HG. Since the subcellular localization of MARCKS is controlled by its phosphorylation, MARCKS acetylation and

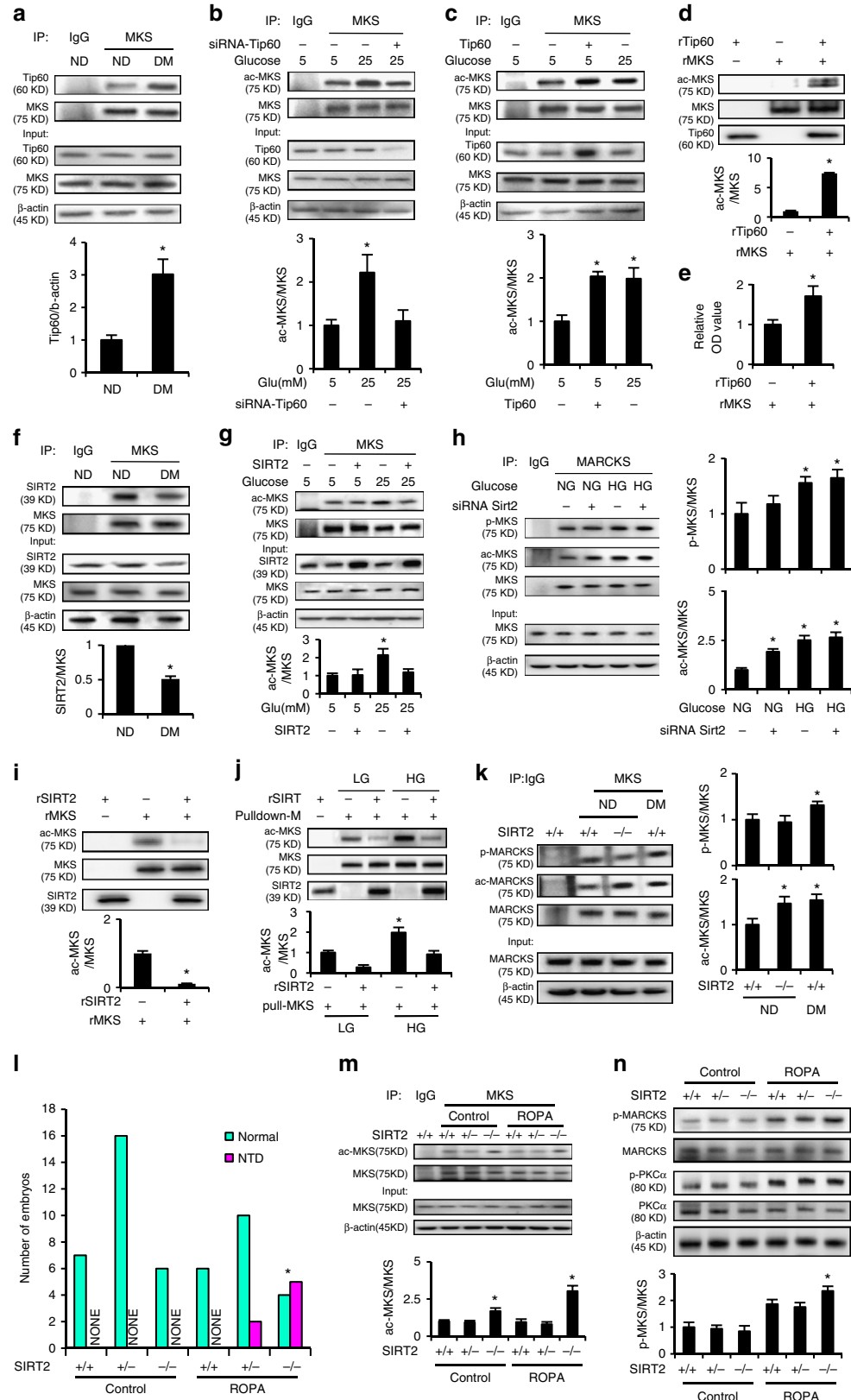

resultant phosphorylation leads to a dissociation from cellular organelle membranes.

Following acetylation, PKC and Protein Phosphatase 2A (PP2A) are responsible for the phosphorylation/depho-sphorylation of MARCKS;[57] however, the enzymes responsible for the acetylation/deacetylation of MARCKS are unknown. This study identified Tip60 as a MARCKS acetyltransferase and SIRT2 as a MARCKS deacetylase. Maternal diabetes in vivo and HG in vitro do not increase Tip60 expression but enhance its activity. In contrast, both SIRT2 expression and activity are down-regulated by maternal diabetes. Further investigation indicated that restoring SIRT2 expression attenuated maternal diabetes-

**Fig. 6** MARCKS is acetylated by Tip60 and deacetylated by SIRT2. **a** Abundance of Tip60 and MARCKS (MKS) in embryos from immunoprecipitation (IP). **b**, **c** Abundance of acetylated MARCKS (ac-MKS) and total MARCKS in cultured cells. **d** Abundance of ac-MARCKS in immunoblotting and (**e**) optical density of ac-MARCKS in a cell free acetylation assay using recombinant MARCKS (rMKS) and rTip60. Abundance of SIRT2 in embryos (**f**) and ac-MARCKS in cultured cells (**g**). Increased acetylation of MARCKS in cultured cells with SIRT2 siRNA knockdown (**h**). ac-MARCKS abundance in a cell free deacetylation assay using rMKS and rSIRT2 (**i**) or using endogenous MARCKS from cultured cells (**j**). Increased acetylation of MARCKS in SIRT2 KO embryos in vivo (**k**). NTDs in SIRT2 KO embryos treated with ROPA (**l**). * Indicates significant difference ($P < 0.05$) in a *Chi*-square test. Increased acetylation (**m**) and phosphorylation (**n**) of MARCKS in SIRT2 KO embryos treated with ROPA. NG: normal glucose (5 mM), HG high glucose (25 mM), ND nondiabetic, DM diabetes mellitus. In **b**, **c**, **d**, **e**, **h**, **i**, experiments were repeated three times. In **a**, **f**, **k**, **m** and **n**, three litters per group (one litter per run) were used. . One way ANOVA with the *Tukey* test was used to analyze the data of panel **b**, **c**, **g**, **h**, **j**, **k**, **m** and **n**. *t* test was used to analyze the data of panel **a**, **d**, **e**, **f** and **i**. * Indicates significant difference ($P < 0.05$) compared to the other groups or the other group

induced NTDs, suggesting that SIRT2-induced MARCKS deacetylation exerts a protective effect on neuroepithelial cells through the inhibition of MARCKS phosphorylation.

In summary, MARCKS acetylation, which is reversibly regulated by Tip60 and SIRT2, is a prerequisite for MARCKS phosphorylation, which is equivalent to loss of function mutations leading to NTDs, and mediates the teratogenicity of maternal diabetes in NTD induction by causing cellular organelle stress. Because MARCKS is broadly involved in an array of neurological disorders[39], identifying avenues, such as acetylation and phosphorylation, to modulate MARCKS activity and restore its functionality is fundamentally important.

## Methods

**Mice**. The procedures used for the animal experiments were approved by the University of Maryland Baltimore Institutional Animal Care and Use Committee. The MARCKS-PD-transgenic (M-Tg)[19] mice (C57BL/6J) were originally obtained from Dr. Perry J. Blackshear at the Signal Transduction Laboratory, National Institute of Environmental Health Sciences. The nestin promoter driven SIRT2 transgenic (SIRT2-Tg) mice were generated on a C57BL/6J background at the Genome Modification Facility, Harvard University. SIRT2 knockout mice were obtained from the Jackson Laboratory (Stock #: 012772, Bar Harber, ME).

**Model of maternal diabetes-induced NTDs**. We[3,11,27,33,55] and other researchers[7,68,69] have used a rodent model of Streptozotocin (STZ)-induced diabetes in research investigating diabetic embryopathy. Eight- to ten-week old female mice were intravenously injected with 65 mg/kg STZ daily into the tail vein over two days to induce diabetes. Diabetes was defined as 12-hour fasting blood glucose concentrations greater than or equal to 14 mM, which usually occurred 3–5 days after the STZ injections. We did not detect any differences in embryonic development between the STZ/insulin-treated and non-STZ-treated mice[6], suggesting that STZ had no residual toxic effect in our animal model. The male M-Tg mice and SIRT2-Tg mice were bred with female diabetic mice. As previously described[3,33,55], the embryos were harvested at E8.75 (2:00 PM at E8.5) for the biochemical and molecular analyses. At E10.5, the embryos were examined under a Leica MZ16F stereomicroscope (Leica, Bannockburn, IL) to identify the NTDs.

**Pathogenic maternal diet treatment**. Eight- to ten-week-old SIRT2 heterozygous (SIRT2[+/−]) female mice were fed a Purina5015 (LabDiet, St. Louis, MO) or control (Purina5001, LabDiet, St. Louis, MO) diet for 1 week prior to mating with the SIRT2 heterozygous (SIRT2[+/−]) male mice. To identify the NTDs, the embryos were examined at E10.5 under a Leica MZ16F stereomicroscope (Leica, Bannockburn, IL).

**Whole embryo cultures**. The SIRT2 heterozygous (SIRT2[+/−]) female and male mice were mated overnight. The day on which a vaginal plug was observed was designated E0.5. At E8.5 (8 AM), mouse embryos with an intact visceral yolk sac were dissected from the uteri into PBS (Invitrogen, La Jolla, CA). Then, three embryos were cultured in 4 ml of culture medium containing 2.66 ml of rat serum, 1.34 ml of Tyrode's salt solution (Cat#: T2397, Sigma, St. Louis, MO), 100 units/mL of penicillin and 100 μg/mL of streptomycin with or without 1 μM ROPA (Resiniferonol-9,13,14-orthophenyl acetate) (Abcam, Cambridge, MA) at 38 °C in a roller bottle system. The embryos were cultured for 36 h under the following conditions: 5% $O_2$—5% $CO_2$—90% $N_2$.

**TUNEL assay**. An ApopTag Red In Situ Apoptosis Detection Kit (Catalog No: S7165, Millipore) was used to detect apoptosis[33]. Ten-micrometers frozen embryonic sections were fixed with 4% PFA in PBS and incubated with TUNEL reaction agents. The percentage of apoptotic cells was obtained in three separate

experiments by dividing the number of TUNEL positive cells by the total number of cells in a microscopic field and then multiplying by 100.

**Cell culture and transfection**. C17.2 mouse neural stem cells, which were originally obtained from the ECACC (European Collection of Cell Culture, Catalogue No.: 07062902, Salisbury, UK), are newborn mouse cerebellar progenitor cells transformed with retroviral v-myc[12,70]. The C17.2 cells were cultured under NG or HG conditions and then transfected with scramble control siRNA or Tip60-siRNA (176470, Life Technology, Carlsbad, CA) using Lipofectamine RNAiMAX (Invitrogen, Carlsbad, CA) according to the protocol from the manufacturer. The C17.2 cells were transfected with pMARCKS-PD, pMARCKS-CA, pMARCKS-DN, pTip60, pSIRT2 and pPKCα-CA using Lipofectamine 2000 (Invitrogen, Carlsbad, CA) according to the protocol from the manufacturer.

**Immunoprecipitation (IP) and immunoblotting**. In total, 300 mg protein from C17.2 cells or five to six embryos pooled per litter, a protease inhibitor cocktail (Sigma-Aldrich, St, Louis, MO), lysis buffer (Cell Signaling Technology, Danvers, MA), Protein A/G Magnetic bead slurry (New England BioLabs, Ipswich, MA), and 1 mg rabbit anti-MARCKS antibody (Santa Cruz Biotechnology, Dallas, TX) were used for IP. The mitochondria were isolated from the C17.2 cells or embryos using a Pierce mitochondria isolation kit (Thermo Fisher Scientific, Waltham, MA). For immunoblotting, equal amounts of protein (30 or 50 mg) from C17.2 cells or one whole embryo per dam were resolved by SDS-PAGE and transferred onto Immobilon-P membranes (Millipore, Billerica, MA). Precision Plus Protein Standards (2 mg; Bio-Rad Laboratories, Hercules, CA) were loaded onto one lane of the gel. The membranes were incubated in 5% nonfat milk for 45 min, followed by incubation with the primary antibodies in 5% nonfat milk for 18 h at 4 °C. The detailed antibody information is provided in Supplementary Table 5. Following the primary antibody incubation, the membranes were exposed to goat anti-rabbit or anti-mouse secondary antibodies. To ensure that equivalent amounts of protein were loaded among the samples, the membranes were stripped and probed with a mouse antibody against β-actin at a dilution of 1:5000 (Abcam, Cambridge, UK) or prohibitin (1:2000) (Calbiochem, Billerica, MA). The signals were detected using a SuperSignal West Femto Maximum Sensitivity Substrate kit (Thermo Fisher Scientific, Waltham, MA). All experiments were repeated three times using independently prepared cell or tissue lysates. All uncropped immunoblotting images were present in Supplementary Fig. 4.

**Immunofluorescence**. Embryos at E8.5 were fixed in 4% paraformaldehyde (PFA) overnight, followed by embedding in optimum cutting temperature medium (OCT) compound (Sakura Finetek, Torrance, CA). Then, 10-μm cryosections were pretreated with citrate buffer and blocked in 5% bovine serum albumin in PBST (0.1% Triton X-100 in PBS) for 1 h. The following antibodies were used as primary antibodies: MARCKS (1:200), TOM20 (1:200), Connexin (1:200), Sox1 (1:200), Tuj1 (1:200) and GFP (1:200). Normal rabbit or mouse IgG at the same dilutions as those of the antibodies were used as controls. After washing with PBS, the sections were incubated with the secondary antibodies. Then, the sections were counterstained with DAPI and mounted with aqueous mounting medium (Sigma, St Louis, MO). The images were captured under a confocal microscope (Carl Zeiss, Obercochen, Germany).

**Hematoxylin-eosin staining**. E10.5 embryos were collected for a morphological examination. First, the embryos were fixed in methcarn (methanol, 60%; chloroform, 30%; and glacial acetic acid, 10%), embedded in paraffin, and cut into 5-μm sections. After deparaffinization and rehydration, all specimens underwent hematoxylin and eosin (H&E) staining using a standard procedure. All embryo sections were photographed under a Nikon Ni-U microscope (Nikon, Tokyo, Japan), and the NTDs were examined.

**Electron microscopy**. The structures of the mitochondria were examined by transmission electron microscopy (EM) at our University EM core facility. Thick sections (1 μM) were cut and visualized at 100× magnifications to identify the neuroepithelium of the E8.5 embryos. Thin sections (80 nm) of the identified

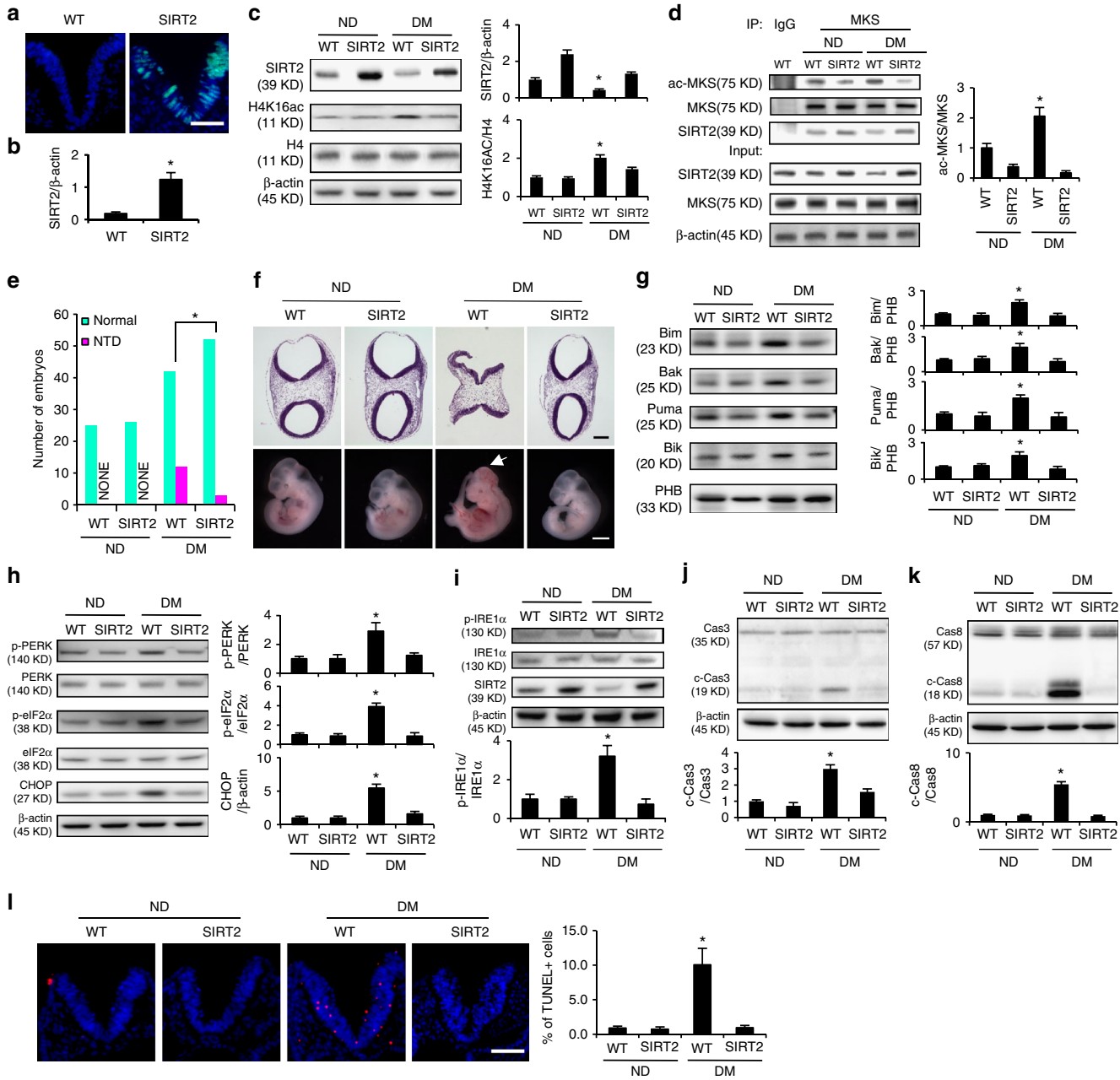

**Fig. 7** SIRT2 overexpression attenuates maternal diabetes-induced neuroepithelial cell apoptosis and ER stress. **a** SIRT2 overexpression (green signals) was observed only in the neuroepithelia of transgenic embryos. Bar indicates 70 μM. **b** mRNA abundance of SIRT2 in transgenic embryos. * Indicates significant difference (*P* < 0.05) in a *t* test. **c** Protein abundance of SIRT2 in transgenic embryos. **d** Abundance of MARCKS acetylation. **e** Numbers of normal and neural tube defect (NTD) embryos. Sample sizes were shown in Supplementary Table 4. **f** Closed or open neural tubes (white arrow) in normal and NTD embryos. Bar in upper and lower panel indicates 600 μm and 1 mm respectively. **g** Mitochondrial abundance of Bim, Bak, Puma and Bik in SIRT2 transgenic embryos. **h**, **i** Abundance of p-PERK, p-eIF2α, CHOP, and IRE1α in SIRT2 transgenic embryos. **j**, **k** Abundance of cleaved caspase 3 and 8 in SIRT2 transgenic embryos. **l** Apoptotic cells labeled as red in embryonic neuroepithelia. Bar indicates 70 μm. ND nondiabetic, DM diabetes mellitus, WT wild type, Cas3 caspase 3, c-Cas3 cleaved caspase 3, Cas8 caspase 8, c-Cas8 cleaved caspase 8. All experiments were repeated three times. Three embryos from three different dams (*N* = 3) were performed in panel **c**, **d**, **g**, **h**, **i**, **j** and **k**. One way ANOVA with the *Tukey* test was used to analyze the data. * Indicates significant difference (*P* < 0.05) compared to the other group or groups

neuroepithelium were cut and viewed under an electron microscope (Joel JEM-1200EX; Tokyo, Japan) at a high-power resolution (6 and 15 K) for the identification of the cellular organelle structures.

**Detection of XBP1 mRNA splicing**. The mRNA of XBP1 was extracted from E8.5 embryos by TRIzol (Invitrogen, Carlsbad, CA) and reverse transcribed to cDNA using a QuantiTect Reverse Transcription Kit (Qiagen, Hilden, Germany). The PCR primers for XBP1 were as follows: forward, 59-GAACCAGGAGTTAAGAAC

ACG-39, and reverse, 59-AGGCAACAGTGTCAGAGTCC-39. If no XBP1 mRNA splicing occurred, a 205-bp band was produced. If XBP1 splicing occurred, a 205-bp band and a 179-bp main band were produced. All primer sequences were listed in Supplementary Table 6.

**Statistical analysis**. The statistical differences in the two group comparisons were determined by Student's *t*-test, and those in the comparisons of more than two groups were determined by one-way ANOVA using SigmaStat 3.5 software. In the

ANOVA analyses, *Tukey* tests were used to estimate the significance of the results. Significant differences between the groups in the NTD incidences were analyzed by the *Chi*-square test. For all immunoblotting experiments, one embryo from one dam in each group was used for one run. Each experiment was repeated three times with three embryos from three different dams each group. For immunoprecipitation experiment, five or six embryos from one dam in one experimental group were combined and used for one run, each experiment was repeated three times with three different litters from different dams per group. All quantification data were indicated as means ± standard derivation.

## Data availability

The authors declare that all data supporting the findings of this study are available within the article and its Supplementary Information Files or from the corresponding author on reasonable request. The data were also deposited in https://figshare.com 7361126 [https://doi.org/10.6084/m9.figshare.7361126.v1].

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

## Acknowledgements

We thank Ms. Hua Li for her technical support and Dr. Julie Rosen at the University of Maryland School of Medicine for her assistance in preparing this manuscript. This work was supported by the NIH NIDDK grants NIH R01DK083243, R01DK101972, R01HL131737, R01HL134368, R01HL139060 and R01DK103024, and an American Diabetes Association Basic Science Award (1-13-BS-220). This research was also supported in part by the Intramural Research Program of the National Institute of Environmental Health Sciences, NIH (D.J.S. and P.J.B.). We thank the support from the Office of Dietary Supplements, National Institute of Health (NIH).

## Author contributions

P.Y., C.X., and X.C. researched data. P.Y. conceived the project, designed the experiments, and wrote the manuscript. E.A.R., M.Z., D.J.S., and P.J.B. participated in data analyses and reviewed the manuscript. J.Z. helped data analysis and reviewed the manuscript.

## Additional information

**Competing interests:** The authors declare no competing interests.

