## [Peer Review File · Nature Communications]

Reviewers' comments:

Reviewer #1 (Remarks to the Author):

This paper investigates the molecular mechanism by which hyperglycemia can lead to abnormal mouse neuroepithelial development, culminating in neural tube closure defects. MARCKS is required for cranial, but not spinal closure, and is convincingly shown to be a key factor in determining susceptibility to NTD, based on its acetylation and phosphorylation. The mitochondrial location of MARCKS allows it to diminish the cellular stress response, whereas this function is compromised after phosphorylation in hyperglycemia. The acetyltransferase Tip60, and acetylase SIRT2, are shown to act upon MARCKS, and a transgenic mouse overexpressing SIRT2 is shown to be protected against hyperglycemia-induced NTDs. This latter part of the study is the most impressive.

While the data are strong in this paper, there are many inaccurate or incorrect statements that need attention.

Title: "Tip60 and Sirtuin 2-regulated MARCKS acetylation and phosphorylation 2 cause diabetic embryopathy". Using the word 'cause' is over-stated here: "are required for" or "participate in" would be a better description of the findings of the paper.

67. "Neuroepithelial cells ... undergo rigorous proliferation, migration and differentiation to acquire the competence for neural plate elevation ...". This is incorrect. Differentiation to form specific cell types such as neurons, floor plate cells, etc occurs only following neural tube closure. The neuroepithelium is undifferentiated during closure.

72. "MARCKS ... may be one of the candidates that exert cellular organelle protective effects in neuroepithelial cells because MARCKS is a membrane-bound protein and primarily resides in neural tissues". This is a non-sequitur. Why should being membrane-bound or primarily "residing" (do you mean "being expressed"?) in neural tissues suggest a candidacy for organelle protection? Please rewrite or remove this sentence.

74. "... deleting the Marcks gene results in NTDs resembling those in diabetic pregnancies". In what way do Marcks-related NTDs particularly resemble those in diabetic pregnancies? Is the resemblance greater than for the > 300 other gene-related NTDs in mice? For example, human diabetic embryopathy includes open spina bifida, but this NTD is not seen in Marcks mutant mice, where only exencephaly is seen. Hence, the resemblance to human diabetic embryopathy is not close, contrary to the authors' statement.

88. "If MARCKS is regulated by acetylation, it is most likely that MARCKS acetylation impacts its phosphorylation." A non-sequitur. Why should this be, without other evidence?

90. "Acetylation was first discovered in histones ..." I think acetylation was known before its action on histones was discovered.

116. "...the osmotic control of glucose, mannitol, had no effect ...". What does this mean? Please rewrite.

122-127. The same point is stated twice in these sentences. It only needs to be said once.

199. "Caspase cleavage-triggered neuroepithelial cell apoptosis is the causal event in diabetes-induced NTDs". What is the evidence for this statement? Apoptosis may occur in DM-induced NTDs, but there

is no direct evidence that cell death prevents closure. For example, which aspect of the closure mechanism does apoptosis prevent? Indeed, this entire paper ignores the actual mechanisms of neural tube closure, and the possible ways in which such mechanisms are disturbed leading to the NTDs.

203. "NTDs can be induced by either delayed or advanced neurogenesis". This is only partially correct. Since neurogenesis follows neural tube closure, it is very unlikely that delayed neurogenesis will prevent neural tube closure. Indeed, reference 37 on the nucleoporin mutant, Nup133, shows that embryos do not survive long enough to close their neural tubes. This does not demonstrate that neurogenesis is required for closure. The effect of premature neurogenesis in inhibiting neural tube closure is, however, well demonstrated in several studies, including reference 38. In consequence, the finding of delayed neurogenesis (TuJ1 +ve cells) in diabetes is of uncertain significance. Please alter the sentences accordingly.

240. "... a transgenic mouse line that overexpressed SIRT2 specifically in the developing neuroepithelium ...". Please state clearly how the transgenic line was targeted to the neuroepithelium.

Fig 1 D-F. Please state which cells these assays were performed in.

There are many errors in English throughout this paper. For example, sentences beginning on the following lines need attention: 35, 37, 63, 78, 83, 109, 110, 168, 213, 257, 258, 273.

Reviewer #2 (Remarks to the Author):

This manuscript by Yang and colleagues assesses the role of acetylation of the MARCKS protein in diabetes-induced neural tube defects (NTDs) using cell culture and mouse models. The authors provide evidence that acetylation of MARCKS leads to its phosphorylation, in turn causing its cytosolic relocalization and consequent mitochondrial/ER dysfunction. They provide evidence that the KAT/KDAC pair TIP60/SIRT2 regulates MARCKS acetylation status. Overexpression of SIRT2 in the developing embryo reduces abnormalities associated with pathogenic maternal diet. Overall, this is a mostly strong manuscript, with a very large number of complementary biochemical, cell culture, and genetic studies. NTDs represent a common but incompletely understood class of fetal anomalies, and these studies may have downstream therapeutic importance. I think it is suitable for publication in Nature Communications, provided the authors can address the following points:

1. The authors show very closely-cropped immunoblots throughout the manuscript. I think they should show whole blots somewhere, likely in the SI.
2. The authors' data on the role of SIRT2 in deacetylating MARCKS is all derived from gain-of-function studies. The authors should perform SIRT2 KD and/or KO to verify that MARCKS acetylation increases in response to SIRT2 loss-of-function. Critically, SIRT2 is a highly active, promiscuous deacetylase, and hence the studies presented do not completely prove that SIRT2 is the physiologic MARCKS deacetylase.
3. Along these lines, the manuscript would be stronger if the authors analyzed the occurrence of NTDs in Sirt2 KO animals, perhaps with and without pathogenic maternal diet. Given the long-term nature of such a study, I do not feel that its absence is a deal-breaker for this paper. However, in its absence I do think the authors need to acknowledge the SIRT2 may not actually be the major MARCKS deacetylase in vivo.

4. Despite their claims, the authors have not really demonstrated mitochondrial dysfunction in Fig. 3, which would require respirometry, ROS measurements, etc. I suggest the term abnormal mitochondrial morphology instead.

Reviewer #3 (Remarks to the Author):

This is a very interesting manuscript describing a novel concept related to MARCKS protein and neural development. The authors are looking at maternal diabetes effects on the developing embryo, and provide evidence that high glucose in vitro and maternal diabetes in vivo lead to MARCKS phosphorylation, which removes MARCKS from binding to mitochondrial and ER membranes and thus removes its protective effects, leading to organellar stress and neural tube defects. Critical to MARCKS phosphorylation is acetylation of MARCKS, which appears to be a prerequisite for phosphorylation. They show nicely that MARCKS is acetylated by Tip60 on lys 165, and deacetylated by SIRT2. Overall, the studies are nicely done and the data appear well-controlled and valid.

Comments and Concerns:

1. The authors are raising an entirely new concept here, that acetylation of MARCKS on a lysine residue is important for its phosphorylation by PKC in neural cells. Since this, to my knowledge, has not been shown before, there are several aspects to this story that need to be addressed. For example, the authors need to speculate on WHY acetylation at Lys 165 (in mice) would make MARCKS more susceptible to phosphorylation. Where does Lys 165 sit on the molecule in relation to the phosphorylation site domain? They state that it is close to the 3 or 4 serine residues that get phosphorylated, but the mechanism of the relationship is not addressed. Is it steric? Does acetylation make the serine sites more accessible?
2. Most importantly, is this a phenomenon restricted to these neural cells or is this an entirely new concept for MARCKS regulation in other tissues and organs?
3. Related to the above, would a MARCKS with the PSD site deleted be acetylated in response to Tip60?
4. The manuscript is filled with typos and errors in English usage, syntax and grammar. Please have the revised MS edited for English.

Responses to Reviewers' comments

Reviewer #1 (Remarks to the Author):

This paper investigates the molecular mechanism by which hyperglycemia can lead to abnormal mouse neuroepithelial development, culminating in neural tube closure defects. MARCKS is required for cranial, but not spinal closure, and is convincingly shown to be a key factor in determining susceptibility to NTD, based on its acetylation and phosphorylation. The mitochondrial location of MARCKS allows it to diminish the cellular stress response, whereas this function is compromised after phosphorylation in hyperglycemia. The acetyltransferase Tip60, and acetylase SIRT2, are shown to act upon MARCKS, and a transgenic mouse overexpressing SIRT2 is shown to be protected against hyperglycemia-induced NTDs. This latter part of the study is the most impressive.

While the data are strong in this paper, there are many inaccurate or incorrect statements that need attention.

Q1: Title: "Tip60 and Sirtuin 2-regulated MARCKS acetylation and phosphorylation 2 cause diabetic embryopathy". Using the word 'cause' is over-stated here: "are required for" or "participate in" would be a better description of the findings of the paper.

A1: Thank you for your comments. We changed the title to "Tip60 and Sirtuin 2-regulated MARCKS acetylation and phosphorylation 2 are required for diabetic embryopathy".

Q2: 67. "Neuroepithelial cells ... undergo rigorous proliferation, migration and differentiation to acquire the competence for neural plate elevation ...". This is incorrect. Differentiation to form specific cell types such as neurons, floor plate cells, etc occurs only following neural tube closure. The neuroepithelium is undifferentiated during closure.

A2: Thank you for your comments. We deleted the word "differentiation".

Q3: 72. "MARCKS ... may be one of the candidates that exert cellular organelle protective effects in neuroepithelial cells because MARCKS is a membrane-bound protein and primarily resides in neural tissues". This is a non-sequitur. Why should being membrane-bound or primarily "residing" (do you mean "being expressed"?) in neural tissues suggest a candidacy for organelle protection? Please rewrite or remove this sentence.

A3: Thank you for your comments. We deleted this sentence.

Q4: 74. "... deleting the Marcks gene results in NTDs resembling those in diabetic pregnancies". In what way do Marcks-related NTDs particularly resemble those in diabetic pregnancies? Is the resemblance greater than for the > 300 other gene-related NTDs in mice? For example, human diabetic embryopathy includes open spina bifida, but this NTD is not seen in Marcks mutant mice, where only exencephaly is seen. Hence, the resemblance to human diabetic embryopathy is not close, contrary to the authors' statement.

A4: Thank you for your comments. Modification was made accordingly. "deleting the Marcks gene results in NTDs, mainly exencephaly."

Q5: 88. "If MARCKS is regulated by acetylation, it is most likely that MARCKS acetylation impacts its phosphorylation." A non-sequitur. Why should this be, without other evidence?

A5: Thank you for your comments. We changed the sentence to “If MARCKS is regulated by acetylation, it is interesting to determine whether MARCKS acetylation impacts its phosphorylation.”

Q6: 90. “Acetylation was first discovered in histones ...” I think acetylation was known before its action on histones was discovered.

A6: Thank you for your comments. We changed it to "Since histone acetylation was first discovered, it became a well-known euchromatin marker in activating gene transcription."

Q7: 116. “...the osmotic control of glucose, mannitol, had no effect ...”. What does this mean? Please rewrite.

A7: Thank you for your comment. When an experiment relative to high glucose was performed, an osmotic control of glucose was usually used to determine that the effect of glucose is from glucose metabolism, not from osmotic pressure. The most common osmotic control of glucose is mannitol. We confirmed here that mannitol had no effect on MARCKS acetylation, but glucose did.

Q8: 122-127. The same point is stated twice in these sentences. It only needs to be said once.

A8: Thank you for your comments. We deleted the sentence (122-124).

Q9: 199. “Caspase cleavage-triggered neuroepithelial cell apoptosis is the causal event in diabetes-induced NTDs”. What is the evidence for this statement? Apoptosis may occur in DM-induced NTDs, but there is no direct evidence that cell death prevents closure. For example, which aspect of the closure mechanism does apoptosis prevent? Indeed, this entire paper ignores the actual mechanisms of neural tube closure, and the possible ways in which such mechanisms are disturbed leading to the NTDs.

A9: We changed the sentence to “Caspase cleavage-triggered neuroepithelial cell apoptosis is observed in diabetes-induced NTDs”. We agree with the reviewer that there is no direct evidence that cell death prevents neural tube closure.

Q10: 203. “NTDs can be induced by either delayed or advanced neurogenesis”. This is only partially correct. Since neurogenesis follows neural tube closure, it is very unlikely that delayed neurogenesis will prevent neural tube closure. Indeed, reference 37 on the nucleoporin mutant, Nup133, shows that embryos do not survive long enough to close their neural tubes. This does not demonstrate that neurogenesis is required for closure. The effect of premature neurogenesis in inhibiting neural tube closure is, however, well demonstrated in several studies, including reference 38. In consequence, the finding of delayed neurogenesis (TuJ1 +ve cells) in diabetes is of uncertain significance. Please alter the sentences accordingly.

A10: We have changed the sentence to “NTDs can be induced by premature neurogenesis¹. However, our previous study demonstrated.....”

Q11: 240. “... a transgenic mouse line that overexpressed SIRT2 specifically in the developing neuroepithelium ...”. Please state clearly how the transgenic line was targeted to the neuroepithelium.

A11: We stated this clearly as “We created a transgenic mouse line in which the SIRT2 transgene was driven by the promoter of the neuroepithelial cell marker nestin². Thus, in this transgenic mouse line, SIRT2 was specifically overexpressed in the developing neuroepithelium (Figure 7A).”

Q12: Fig 1 D-F. Please state which cells these assays were performed in.

A12: We have stated the cells as the C17.2 cells.

Q13: There are many errors in English throughout this paper. For example, sentences beginning on the following lines need attention: 35, 37, 63, 78, 83, 109, 110, 168, 213, 257, 258, 273.

A13: Thank you for your comments and we have made corrections accordingly. Please see the corresponding page numbers.

Line 35: MARCKS is required for neural tube closure but its regulation and function are elusive. (In Page 36).

Line 37: MARCKS acetylation is a prerequisite for its phosphorylation. (In Page 38).

Line 63: Failed neural tube closure leads to neural tube defects (NTDs), severe structural birth defects affecting offspring mortality and morbidity. (In Page 68).

Line 78: PKCs are cellular stress kinases because deleting Prkcd genes abolishes maternal diabetes-induced cellular organelle stress in neurulation stage embryos. (In Page 80).

Line 83: The crosstalk between acetylation and phosphorylation has been revealed in transcription factors. (In Page 85).

Line 109: High glucose-induced MARCKS acetylation in K165, a prerequisite for its phosphorylation (In Page 129).

Line 110: Hyperglycemia of maternal diabetes enhances histone acetylation in the developing embryo (In Page 130).

Line 168: Endoplasmic reticulum (ER) stress is another form of cellular organelle stress involved the pathogenesis of diabetic embryopathy (In Page 193).

Line 213: MARCKS acetylation was prerequisite for its phosphorylation (In Page 241).

Line 257: MARCKS is one of the more than 300 genes whose deletions cause NTDs (In Page 288).

Line 258: There is limited information about these genes function during neurulation.
(In Page 292).

Line 273: Abnormalities due to MARCKS deficiency can be rescued, at least partially, by two types of MARCKS mutants: the phosphorylation-dead site mutant, MARCKS-PD (serines to asparagines), and the non-myristoylated MARCKS mutant.
(In Page 307).

Reviewer #2 (Remarks to the Author):

This manuscript by Yang and colleagues assesses the role of acetylation of the MARCKS protein in diabetes-induced neural tube defects (NTDs) using cell culture and mouse models. The authors provide evidence that acetylation of MARCKS leads to its phosphorylation, in turn causing its cytosolic relocation and consequent mitochondrial/ER dysfunction. They provide evidence that the KAT/KDAC pair TIP60/SIRT2 regulates MARCKS acetylation status. Overexpression of SIRT2 in the developing embryo reduces abnormalities associated with pathogenic maternal diet. Overall, this is a mostly strong manuscript, with a very large number of complementary biochemical, cell culture, and genetic studies. NTDs represent a common but incompletely understood class of fetal anomalies, and these studies may have downstream therapeutic importance. I think it is suitable for publication in Nature Communications, provided the authors can address the following points:

Q1. The authors show very closely-cropped immunoblots throughout the manuscript. I think they should show whole blots someplace, likely in the SI.

A1. Thank you for your comments. All whole blots were attached in SI.

Q2. The authors' data on the role of SIRT2 in deacetylating MARCKS is all derived from gain-of-function studies. The authors should perform SIRT2 KD and/or KO to verify that MARCKS acetylation increases in response to SIRT2 loss-of-function. Critically, SIRT2 is a highly active, promiscuous deacetylase, and hence the studies presented do not completely prove that SIRT2 is the physiological MARCKS deacetylase.

A2. We have obtained the SIRT2 knockout (KO) mice and performed the suggested experiments. We found that deleting the *Sirt2* gene *in vivo* increased MARCKS acetylation (**Fig.6K**). SIRT2 knockdown *in vitro* also increased MARCKS acetylation (**Fig.6H**).

Q3. Along these lines, the manuscript would be stronger if the authors analyzed the occurrence of NTDs in *Sirt2* KO animals, perhaps with and without pathogenic maternal diet. Given the long-term nature of such a study, I do not feel that its absence is a deal-breaker for this paper. However, in its absence I do think the authors need to acknowledge the SIRT2 may not actually be the major MARCKS deacetylase *in vivo*.

A3. SIRT2 KO embryos did not exhibit any NTDs under nondiabetic conditions, suggesting that MARCKS acetylation only is not sufficient to induce NTDs. To determine whether a pathogenic maternal diet will lead to NTD formation in SIRT2 KO embryos, SIRT2 heterozygous (*SIRT2*^{+/-}) female mice were fed with a pathogenic maternal diet³, Purina5015 (LabDiet, St.Louis, MO), and mated with *SIRT2*^{+/-} male mice. Wild-Type (WT), *SIRT2*^{+/-} and *SIRT2*^{-/-} embryos all developed normally without NTDs (**Table S2**), suggesting that even though SIRT2 KO increases MARCKS acetylation, the pathogenic maternal diet might not be able to induce MARCKS phosphorylation, a requirement for NTD formation. Because SIRT2 down-regulation and PKC α

activation are concomitantly present in diabetic embryopathy^{2,4}, we tested whether PKC α activation in combination with SIRT2 KO would result in NTDs. WT, SIRT2^{+/-} and SIRT2^{-/-} embryos were cultured with or without ROPA, a PKC α activator. While ROPA treatment did not induce NTDs in WT embryos, it induced 16.7% and 55.6% NTDs in SIRT2^{+/-} and SIRT2^{-/-} embryos, respectively (**Fig.6L**, **Table S3**). Increased MARCKS acetylation were observed in SIRT2^{-/-} embryos (**Fig.6M**), and ROPA treatment increased MARCKS phosphorylation in SIRT2^{-/-} embryos compared to that in WT embryos (**Fig.6N**).

These evidence support our hypothesis that both MARCKS acetylation and phosphorylation are required for NTD induction, and MARCKS acetylation renders its phosphorylation by PKC α possible.

Q4. Despite their claims, the authors have not really demonstrated mitochondrial dysfunction in Fig. 3, which would require respirometry, ROS measurements, etc. I suggest the term abnormal mitochondrial morphology instead.

A4: We changed the term to abnormal mitochondrial morphology.

Reviewer #3 (Remarks to the Author):

This is a very interesting manuscript describing a novel concept related to MARCKS protein and neural development. The authors are looking at maternal diabetes effects on the developing embryo, and provide evidence that high glucose in vitro and maternal diabetes in vivo lead to MARCKS phosphorylation, which removes MARCKS from binding to mitochondrial and ER membranes and thus removes its protective effects, leading to organelle stress and neural tube defects. Critical to MARCKS phosphorylation is acetylation of MARCKS, which appears to be a prerequisite for phosphorylation. They show nicely that MARCKS is acetylated by Tip60 on lys 165, and deacetylated by SIRT2.

Overall, the studies are nicely done and the data appear well-controlled and valid.

Comments and Concerns:

Q1. The authors are raising an entirely new concept here, that acetylation of MARCKS on a lysine residue is important for its phosphorylation by PKC in neural cells. Since this, to my knowledge, has not been shown before, there are several aspects to this story that need to be addressed. For example, the authors need to speculate on WHY acetylation at Lys 165 (in mice) would make MARCKS more susceptible to phosphorylation. Where does Lys 165 sit on the molecule in relation to the phosphorylation site domain? They state that it is close to the 3 or 4 serine residues that get phosphorylated, but the mechanism of the relationship is not addressed. Is it steric? Does acetylation make the serine sites more accessible?

A1.

MARCKS on lysine residue 165 (K165) is important for its phosphorylation by PKC in neural cells. The phosphorylation sites in MARKCS' PSD (Phosphorylation site domain) are Serine 152 (S152), S156, S160 and S163. The acetylation residue K165 sits adjacent to these four serine residues. K165 acetylation brings in a negative charge that may neutralize the positive charge in nearby amino acid residues leading to more open structure of the MARCKS protein, particularly in the PSD. This structural change may make PKC more accessible to the four serine sites and potentiate them being phosphorylated. However, a direct investigation on the physical function of acetylation on phosphorylation is very challenging. Future studies may explore these structural changes in the MARCKS protein upon acetylation.

We included this statement in Lines 392-399 in the Discussion.

Q2. Most importantly, is this a phenomenon restricted to these neural cells or is this an entirely new concept for MARCKS regulation in other tissues and organs?

A2. MARCKS is mainly expressed in neural tissues. Thus, key studies were conducted in the developing neuroepithelium. However, in some of the biochemical studies, we used whole embryos to make the measurement feasible (the amount of neural tissues is very limited), and the findings in these studies are identical to those in neuroepithelium-specific studies. Therefore, we predict that the discovered new mechanism underlying MARCKS regulation is also applicable to other tissues and organs.

Q3. Related to the above, would a MARCKS with the PSD site deleted be acetylated in response to Tip60?

A3. The acetylation site of K165 is inside of the PSD of MARCKS and two amino acids away from phosphorylation site of S163. If the PSD is deleted, the acetylation site is also missing.

Q4. The manuscript is filled with typos and errors in English usage, syntax and grammar. Please have the revised MS edited for English.

A4. We have a native English speaker edit the manuscript.

Reference:

- 1 Ishibashi, M. *et al.* Targeted disruption of mammalian hairy and Enhancer of split homolog-1 (HES-1) leads to up-regulation of neural helix-loop-helix factors, premature neurogenesis, and severe neural tube defects. *Genes Dev* **9**, 3136-3148 (1995).
- 2 Wang, F. *et al.* Protein kinase C-alpha suppresses autophagy and induces neural tube defects via miR-129-2 in diabetic pregnancy. *Nat Commun* **8**, 15182, doi:10.1038/ncomms15182 (2017).
- 3 Kappen, C., Kruger, C., MacGowan, J. & Salbaum, J. M. Maternal diet modulates the risk for neural tube defects in a mouse model of diabetic pregnancy. *Reprod Toxicol* **31**, 41-49, doi:10.1016/j.reprotox.2010.09.002 (2011).
- 4 Yu, J., Wu, Y. & Yang, P. High glucose-induced oxidative stress represses sirtuin deacetylase expression and increases histone acetylation leading to neural tube defects. *J Neurochem* **137**, 371-383, doi:10.1111/jnc.13587 (2016).

REVIEWERS' COMMENTS:

Reviewer #1 (Remarks to the Author):

In the revised version of this manuscript, the authors have answered all my points in a satisfactory way. I have a small number of additional comments that need attention:

Line 202. "NTDs can be induced by premature neurogenesis". This point is not well expressed. I suggest: "NTDs can result from genetic insults that cause premature neurogenesis".

Lines 240-241. "Pathogenic diet". The Methods simply names the diet: 'Purina5015', but no details are given of the content of the diet nor why it is pathogenic. More explanation and detail is needed.

Lines 264-267. Citation of Figure 6 E,F is incorrect. Should be Figure 7 E,F.

The standard of English continues to be poor in many places, and particularly affects parts of the text that are newly written in the revised version. It looks like the original manuscript was proof-read for English, but not the newly written sentences. For example:

Lines 78-80: "These evidence suggest that PKC triggers cellular stress by suppressing the protective effects of MARCKS on cellular organelle." There are 2 grammatical errors in this sentence alone.

Lines 207-208. "In contrast, TuJ1 expressing neurons were appeared in the floor plate ..." English needs attention.

Line 324: "... and a lack of a key mitochondrial redox genes result in NTDs". Poor English.

Reviewer #2 (Remarks to the Author):

The changes made to the manuscript by Yang and colleagues have greatly strengthened it. The new data on SIRT2 KO mice are particularly compelling. The piece should be of broad interest to the readership of Nature Communications. I do have a couple of remaining, very minor comments. First, the writing in the introduction is still somewhat rough in places, though much improved from the initial submission. The authors switch back and forth between present and past tense, and this section could use further copy-editing for style. Second, as I noted in my previous comments, aberrant mitochondrial morphology is not itself firm evidence for mitochondrial dysfunction, nor is increased expression of pro-apoptotic proteins. The phrase mitochondrial dysfunction still appears at line 159, and in the legend to Fig. 3. If the authors wish to substantiate the presence of dysfunctional mitochondria, they would need to perform additional, functional assays.

Reviewer #3 (Remarks to the Author):

The authors have responded appropriately to my previous comments and concerns

Responses to the reviewers' concerns

Reviewer #1 (Remarks to the Author):

In the revised version of this manuscript, the authors have answered all my points in a satisfactory way. I have a small number of additional comments that need attention:

Line 202. NTDs can be induced by premature neurogenesis. This point is not well expressed. I suggest: NTDs can result from genetic insults that cause premature neurogenesis.

Response: Thank you for your comments. We have changed the text as suggested.

Lines 240-241. Pathogenic diet?. The Methods simply names the diet: Purina5015, but no details are given of the content of the diet nor why it is pathogenic. More explanation and detail is needed.

Response: Thank you for your comments. Because the diet of Purina5015 increases the NTD rate of mouse embryos up to 3 times higher than the control diet in the STZ-induced diabetes mouse model, we tested whether Purina5015 diet could be used as pathogenic diet in our model. This was suggested by Reviewer 2.

Lines 264-267. Citation of Figure 6 E,F is incorrect. Should be Figure 7 E,F.

Response: Thank you for your comments. Corrected.

The standard of English continues to be poor in many places, and particularly affects parts of the text that are newly written in the revised version. It looks like the original manuscript was proof-read for English, but not the newly written sentences. For example:

Lines 78-80: These evidence suggest that PKC triggers cellular stress by suppressing the protective effects of MARCKS on cellular organelle. There are 2 grammatical errors in this sentence alone.

Response: corrected accordingly.

Lines 207-208. ???In contrast, TuJ1 expressing neurons were appeared in the floor plate. English needs attention.

Response: corrected accordingly. "In contrast, TuJ1 expressing neurons appeared in the floor plate."

Line 324: and a lack of a key mitochondrial redox genes result in NTDs. Poor English.

Response: corrected accordingly. "and lack of a key mitochondrial redox gene result in NTDs"

Reviewer #2 (Remarks to the Author):

The changes made to the manuscript by Yang and colleagues have greatly strengthened it. The new data on SIRT2 KO mice are particularly compelling. The piece should be of broad interest to the readership of Nature Communications. I do have a couple of remaining, very minor comments. First, the writing in the introduction is still somewhat rough in places, though much improved from the initial submission. The

authors switch back and forth between present and past tense, and this section could use further copy-editing for style. Second, as I noted in my previous comments, aberrant mitochondrial morphology is not itself firm evidence for mitochondrial dysfunction, nor is increased expression of pro-apoptotic proteins. The phrase mitochondrial dysfunction still appears at line 159, and in the legend to Fig. 3. If the authors wish to substantiate the presence of dysfunctional mitochondria, they would need to perform additional, functional assays.

Response: Thank you for your comments. The manuscript had been edited by the Nature Research Editing Service (<http://bit.ly/NRES-NC>). "mitochondrial dysfunction" was changed to "mitochondrial abnormalities".